# Data-driven Optimal Filtering for Linear Systems with Unknown Noise Covariances

**Shahriar Talebi**[1,2]    **Amirhossein Taghvaei**[1]    **Mehran Mesbahi**[1]

[1]University of Washington, Seattle, WA, 98105
[2]Harvard University, Cambridge, MA, 02138

`talebi@seas.harvard.edu`    `amirtag@uw.edu`    `mesbahi@uw.edu`

## Abstract

This paper examines learning the optimal filtering policy, known as the Kalman gain, for a linear system with unknown noise covariance matrices using noisy output data. The learning problem is formulated as a stochastic policy optimization problem, aiming to minimize the output prediction error. This formulation provides a direct bridge between data-driven optimal control and, its dual, optimal filtering. Our contributions are twofold. Firstly, we conduct a thorough convergence analysis of the stochastic gradient descent algorithm, adopted for the filtering problem, accounting for biased gradients and stability constraints. Secondly, we carefully leverage a combination of tools from linear system theory and high-dimensional statistics to derive bias-variance error bounds that scale logarithmically with problem dimension, and, in contrast to subspace methods, the length of output trajectories only affects the bias term.

## 1 Introduction

The duality of control and estimation plays a crucial role in system theory, linking two distinct synthesis problems [1, 2, 3, 4, 5, 6]. This duality is an effective bridge between two distinct disciplines, facilitating development of theoretical and computational techniques in one domain and then adopting them for use in the other. For example, the stability proof of the Kalman filter relies on the stabilizing characteristic of the optimal feedback gain in the dual Linear Quadratic Regulator (LQR) optimal control problem [7, Ch. 9]. In this paper, we leverage this duality to learn optimal filtering policies using recent advances in data-driven algorithms for optimal control.

We consider the estimation problem for a system with a known linear dynamic and observation model, but unknown process and measurement noise covariances. Our objective is to learn the optimal steady-state Kalman gain using a training dataset comprising independent realizations of the observation signal. This problem has a rich history in system theory, often explored within the context of adaptive Kalman filtering [8, 9, 10, 11, 12, 13]. A comprehensive summary of four solution approaches to this problem can be found in the classical reference [9]. These approaches include Bayesian inference [14, 15, 16], Maximum likelihood [17, 18], covariance matching [12], and innovation correlation methods [8, 10]. While Bayesian and maximum likelihood approaches are computationally intensive, and covariance matching introduces biases in practice, the innovation correlation-based approaches have gained popularity and have been the subject of recent research [19, 20, 21]. For an excellent survey on this topic, refer to the article [22]. However, it is important to note that these approaches often lack non-asymptotic guarantees and heavily depend on statistical assumptions about the underlying model.

In the realm of optimal control, significant progress has been made in the development of data-driven synthesis methods. Notably, recent advances have focused on the adoption of first-order methods for state-feedback LQR problems [23, 24]. The direct optimization of policies from a gradient-dominant

perspective has first proven in [25] to be remarkably effective with global convergence despite non-convex optimization landscape. It has been demonstrated that despite the non-convex nature of the cost function, when expressed directly in terms of the policy, first-order methods exhibit global convergence to the optimal policy. Building upon this line of work, the use of first-order methods for policy optimization has been explored in variants of the LQR problem. These include Output-feedback Linear Quadratic Regulators (OLQR) [26], model-free setup [27], risk-constrained setup [28], Linear Quadratic Gaussian (LQG) [29], and most recently, Riemannian constrained LQR [30]. These investigations have expanded the scope of data-driven optimal control, demonstrating the versatility and applicability of first-order methods for a wide range of synthesis problems.

The objective of this paper is to provide fresh insights into the classical estimation problem by leveraging the duality between control and estimation and incorporating recent advances in data-driven optimal control. Specifically, building on the fundamental connection between the optimal mean-squared error estimation problem and the LQR problem (Prop. 1), we reformulate determining the optimal Kalman gain as a problem of synthesizing an optimal policy for the adjoint system, under conditions that differ from those explored in the existing literature (see (10) and Remark 3). Upon utilizing this relationship, we propose a Stochastic Gradient Descent (SGD) algorithm for learning the optimal Kalman gain, accompanied by novel non-asymptotic error guarantees in presence of biased gradient and stability constraint. Our approach opens up promising avenues for addressing the estimation problem with robust and efficient data-driven techniques. The following is an informal statement of our main results (combination of Thm. 1 and Thm. 2), and missing proofs appear in the supplementary materials.

**Theorem 3** (Informal). *Suppose the system is observable and both dynamic and measurement noise are bounded. Then, with high probability, direct policy updates using stochastic gradient descent with small stepsize converges* linearly *and* globally *(from any initial stabilizing policy) to the optimal steady-state Kalman gain.*

More recently, the problem of learning the Kalman gain has been considered from a system identification perspective, for completely *unknown* linear systems [31, 32, 33, 34]. In [32] and [33], subspace system identification methods are used to obtain error bounds for learning the Markov parameters of the model over a time horizon and establish logarithmic regret guarantee for output prediction error. Due to the inherent difficulty of learning a completely unknown stochastic system from partial observations, subspace methods assume marginal *stability of the unknown system*, and lead to sub-optimal sample complexity bounds that grow with the number of Markov parameters, instead of the number of unknowns [35, pp. 14]. Alternatively, [34] considers minimizing the output prediction error and introduces a model-free policy gradient approach, under the same stability assumptions, that achieves *sublinear convergence rate*. This paper provides a middle ground between completely known and completely unknown systems, for a learning scenario that not only has relevant practical implications, but also utilizes the duality relationship to LQR to establish *linear convergence rates* even for *unstable systems* as long as they are *observable*. See the Appendix A for the discussion of additional related works [36, 37, 38].

## 2 Background and Problem Formulation

Herein, first we propose the model setup in detail and discuss the Kalman filter as the estimation strategy. Consider the discrete-time filtering problem given by the stochastic difference equations,

$$x(t+1) = Ax(t) + \xi(t), \quad \text{and} \quad y(t) = Hx(t) + \omega(t), \tag{1}$$

where $x(t) \in \mathbb{R}^n$ is the state of the system, $y(t) \in \mathbb{R}^m$ is the observation signal, and $\{\xi(t)\}_{t \in \mathbb{Z}}$ and $\{\omega(t)\}_{t \in \mathbb{Z}}$ are the uncorrelated zero-mean random vectors, that represent the process and measurement noise respectively, with the following covariances,

$$\mathbb{E}\left[\xi(t)\xi(t)^{\mathsf{T}}\right] = Q \in \mathbb{R}^{n \times n}, \quad \mathbb{E}\left[\omega(t)\omega(t)^{\mathsf{T}}\right] = R \in \mathbb{R}^{m \times m},$$

for some positive (semi-)definite matrices $Q, R \succeq 0$. Let $m_0$ and $P_0 \succeq 0$ denote the mean and covariance of the initial condition $x_0$.

In the filtering setup, the state $x(t)$ is hidden, and the objective is to estimate it given the history of the observation signal $\mathcal{Y}(t) = \{y(0), y(1), \ldots, y(t-1)\}$. The best linear mean-squared error (MSE)

estimate of $x(t)$ is defined according to

$$\hat{x}(t) = \underset{\hat{x} \in \mathcal{L}(\mathcal{Y}(t))}{\arg\min} \; \mathbb{E}\left[\|x(t) - \hat{x}\|^2\right] \tag{2}$$

where $\mathcal{L}(\mathcal{Y}(t))$ denotes the space of all linear functions of the history of the observation signal $\mathcal{Y}(t)$. If the model parameters $(A, H, Q, R)$ are known, the optimal MSE estimate $\hat{x}(t)$ can be recursively computed by the Kalman filter algorithm [2]:

$$\hat{x}(t+1) = A\hat{x}(t) + L(t)(y(t) - H\hat{x}(t)), \quad \hat{x}(0) = m_0, \tag{3a}$$

$$P(t+1) = AP(t)A^\intercal + Q - AP(t)H^\intercal S(t)^{-1}HP(T)A^\intercal, \quad P(0) = P_0, \tag{3b}$$

where $S(t) = HP(t)H^\intercal + R$, $L(t) := AP(t)H^\intercal S(t)^{-1}$ is the Kalman gain, and $P(t) := \mathbb{E}[(x(t) - \hat{x}(t))(x(t) - \hat{x}(t))^\intercal]$ is the error covariance matrix.

**Assumption 1.** *The pair $(A, H)$ is detectable, and the pair $(A, Q^{\frac{1}{2}})$ is stabilizable, where $Q^{\frac{1}{2}}$ is the unique positive semidefinite square root of $Q$.*

Under this assumption, the error covariance $P(t)$ converges to a steady-state value $P_\infty$, resulting in a unique steady-state Kalman gain $L_\infty = AP_\infty H^\intercal (HP_\infty H^\intercal + R)^{-1}$[39, 40]. It is common to evaluate the steady-state Kalman gain $L_\infty$ offline and use it, instead of $L(t)$, to update the estimate in real-time. Furthermore, we note that the assumption of uncorrelated random vectors is sufficient to establish that Kalman filter provides the best *linear* estimate of the states given the observations for minimizing the MSE criterion [2, Theorem 2].

## 2.1 Learning problem

Now, we describe our learning setup: 1) The system matrices $A$ and $H$ are known, but the process and the measurement noise covariances, $Q$ and $R$, are *not* available. 2) We have access to an oracle that generates independent realizations of the observation signal for given length $T$: $\{y(t)\}_{t=0}^T$. However, ground-truth measurements of the state $x(t)$ is *not* available.

*Remark* 1. Our proposed learning setup arises in various important engineering applications where merely approximate or reduced-order linear models are available due to difficulty in analytically capturing the effect of complex dynamics or disturbances, hence represented by noise with unknown covariance matrices. Additionally, the system identification procedure often occurs through the application of physical principles and collection of data from experiments in a controlled environment (e.g., in a wind tunnel). However, identifying the noise covariance matrices strongly depends on the operating environment which might be significantly different than the experimental setup. Therefore, it is common engineering practice to use the learned system matrices and tune the Kalman gain to improve the estimation error. We refer to [41] for the application of this procedure for gust load alleviation in wings and [42] for estimation in chemical reactor models. We also emphasize that this learning setup has a rich history in adaptive filtering with numerous references with a recent survey on this topic [22]. As part of our future research, we will carry-out a robustness analysis, similar to its LQR dual counterpart [43, 44], to study the effect of the error in system matrices on the learning performance.

Inspired by the structure of the Kalman filter, our goal is to learn the steady-state Kalman gain $L_\infty$ from the data described in the learning setup:

> Given: independent random realizations of $\{y(0), \dots, y(T)\}$ with the parameters $A, H$
> Learn: steady-state Kalman gain $L_\infty$

For that, we formulate the learning problem as a stochastic optimization described next.

## 2.2 Stochastic optimization formulation

Define $\hat{x}_L(T)$ to be the estimate given by the Kalman filter at time $T$ realized by the constant gain $L$. Rolling out the update law (3a) for $t = 0$ to $t = T - 1$, and replacing $L(t)$ with $L$, leads to the following expression for the estimate $\hat{x}_L(T)$ as a function of $L$,

$$\hat{x}_L(T) = A_L^T m_0 + \sum_{t=0}^{T-1} A_L^{T-t-1} Ly(t), \tag{4}$$

where $A_L := A - LH$. Note that evaluating this estimate does not require knowledge of $Q$ or $R$. However, it is not possible to directly aim to learn the gain $L$ by minimizing the MSE (2) because the ground-truth measurement of the state $x(T)$ is not available. Instead, we propose to minimize the MSE in predicting the observation $y(T)$ as a surrogate objective function:[1]

$$\min_L \ J_T^{\text{est}}(L) := \mathbb{E}\left[\|y(T) - \hat{y}_L(T)\|^2\right] \tag{5}$$

where $\hat{y}_L(T) := H\hat{x}_L(T)$. Note that while the objective function involves finite time horizon $T$, our goal is to learn the steady-state Kalman gain $L_\infty$.

The justification for using the surrogate objective function in (5) instead of MSE error (2) lies in the detectability assumption 1. Detectability implies that all unobservable states are stable; in other words, their impact on the output signal vanishes quickly—depending on their stability time constant.

Numerically, the optimization problem (5) falls into the category of stochastic optimization and can be solved by algorithms such as Stochastic Gradient Descent (SGD). Such an algorithm would need access to independent realizations of the observation signal which are available. Theoretically however, it is not yet clear if this optimization problem is well-posed and admits a unique minimizer. This is the subject of following section where certain properties of the objective function, such as its gradient dominance and smoothness, are established. These theoretical results are then used to analyze first-order optimization algorithms and provide stability guarantees of the estimation policy iterates. The results are based on the duality relationship between estimation and control that is presented next.

## 3   Estimation-Control Duality Relationship

The stochastic optimization problem (5) is related to an LQR problem through the application of the classical duality relationship between estimation and control [45, Ch.7.5]. In order to do so, we introduce the adjoint system (dual to (1)) according to:

$$z(t) = A^\mathsf{T} z(t+1) - H^\mathsf{T} u(t+1), \quad z(T) = a \tag{6}$$

where $z(t) \in \mathbb{R}^n$ is the adjoint state and $\mathcal{U}(T) := \{u(1), \ldots, u(T)\} \in \mathbb{R}^{mT}$ are the control variables (dual to the observation signal $\mathcal{Y}(T)$). The adjoint state is initialized at $z(T) = a \in \mathbb{R}^n$ and simulated *backward in time* starting with $t = T - 1$. We introduce an LQR cost for the adjoint system:

$$J_T^{\text{LQR}}(a, \mathcal{U}_T) := z^\mathsf{T}(0) P_0 z(0) + \sum_{t=1}^{T} \left[z^\mathsf{T}(t) Q z(t) + u^\mathsf{T}(t) R u(t)\right], \tag{7}$$

and formalize a relationship between linear estimation policies for the system (1) and linear control policies for the adjoint system (6). A linear estimation policy takes the observation history $\mathcal{Y}_T \in \mathbb{R}^{mT}$ and outputs an estimate $\hat{x}_\mathcal{L}(T) := \mathcal{L}(\mathcal{Y}_T)$ where $\mathcal{L} : \mathbb{R}^{mT} \to \mathbb{R}^n$ is a linear map. The adjoint of this linear map, denoted by $\mathcal{L}^\dagger : \mathbb{R}^n \to \mathbb{R}^{mT}$, is used to define a control policy for the adjoint system (6) which takes the initial condition $a \in \mathbb{R}^n$ and outputs the control signal $\mathcal{U}_{\mathcal{L}^\dagger} = \mathcal{L}^\dagger(a)$; i.e.,

$$\{y(0), \ldots, y(T-1)\} \xrightarrow{\mathcal{L}} \hat{x}_\mathcal{L}(T) \qquad \text{and} \qquad \{u(1), \ldots, u(T)\} \xleftarrow{\mathcal{L}^\dagger} a.$$

The duality relationship between optimal MSE estimation and LQR control is summarized in the following proposition. The proof is presented in the supplementary material.

**Proposition 1.** *Consider the estimation problem for the system* (1) *and the LQR problem* (7) *subject to the adjoint dynamics* (6). *For any linear estimation policy $\hat{x}_\mathcal{L}(T) = \mathcal{L}(\mathcal{Y}_T)$, and for any $a \in \mathbb{R}^n$, we have the identity*

$$\mathbb{E}\left[|a^\mathsf{T} x(T) - a^\mathsf{T} \hat{x}_\mathcal{L}(T)|^2\right] = J_T^{LQR}(a, \mathcal{U}_{\mathcal{L}^\dagger}(T)), \tag{8}$$

*where $\mathcal{U}_{\mathcal{L}^\dagger}(T) = \mathcal{L}^\dagger(a)$. In particular, for a Kalman filter with constant gain $L$, the output prediction error* (5)

$$\mathbb{E}\left[\|y(T) - \hat{y}_L(T)\|^2\right] = \sum_{i=1}^{m} J_T^{LQR}(H_i, \mathcal{U}_{L^\mathsf{T}}(T)) + \text{tr}\left[R\right], \tag{9}$$

*where $\mathcal{U}_{L^\mathsf{T}}(T) = \{L^\mathsf{T} z(1), L^\mathsf{T} z(2), \ldots, L^\mathsf{T} z(T)\}$, i.e., the feedback control policy with constant gain $L^\mathsf{T}$, and $H_i^\mathsf{T} \in \mathbb{R}^n$ is the i-th row of the $m \times n$ matrix $H$ for $i = 1, \ldots, m$.*

---

[1]The expectation in (5) is taken over all the random variables; consisting of the initial state $x_0$, dynamic noise $\xi(t)$, and measurement noise $\omega(t)$ for $t = 0, \cdots, T$. A conditional expectation is not necessary as the estimate is constrained to be measurable with respect to the history of observation.

*Remark* 2. The duality is also true in the continuous-time setting where the estimation problem is related to a continuous-time LQR. Recent extensions to the nonlinear setting appears in [46] with a comprehensive study in [47]. This duality is distinct from the maximum likelihood approach which involves an optimal control problem over the original dynamics instead of the adjoint system [5].

## 3.1 Duality in steady-state regime

Using the duality relationship (9), the MSE in prediction (5) is expressed as:

$$J_T^{\text{est}}(L) = \text{tr}\left[X_T(L)H^\intercal H\right] + \text{tr}\left[R\right],$$

where $X_T(L) := A_L^T P_0 (A_L^\intercal)^T + \sum_{t=0}^{T-1} A_L^t (Q + LRL^\intercal)(A_L^\intercal)^t$. Define the set of Schur stabilizing gains

$$\mathcal{S} := \{L \in \mathbb{R}^{n \times m} : \rho(A_L) < 1\}.$$

For any $L \in \mathcal{S}$, in the steady-state, the mean-squared prediction error assumes the form,

$$\lim_{T \to \infty} J_T^{\text{est}}(L) = \text{tr}\left[X_\infty(L)H^\intercal H\right] + \text{tr}\left[R\right],$$

where $X_\infty(L) := \lim_{T \to \infty} X_T(L)$ and coincides with the unique solution $X$ of the discrete Lyapunov equation $X = A_L X A_L^\intercal + Q + LRL^\intercal$ (existence of unique solution follows from $\rho(A_L) < 1$). Given the steady-state limit, we formally analyze the following constrained optimization problem:

$$\min_{L \in \mathcal{S}} \leftarrow J(L) := \text{tr}\left[X_{(L)} H^\intercal H\right], \quad \text{s.t.} \quad X_{(L)} = A_L X_{(L)} A_L^\intercal + Q + LRL^\intercal. \tag{10}$$

*Remark* 3. Note that the latter problem is technically the dual of the optimal LQR problem as formulated in [23] by relating $A \leftrightarrow A^\intercal$, $-H \leftrightarrow B^\intercal$, $L \leftrightarrow K^\intercal$, and $H^\intercal H \leftrightarrow \Sigma$. However, the main difference here is that the product $H^\intercal H$ may *not* be positive definite, for example, due to rank deficiency in $H$ specially when $m < n$ (whereas $\underline{\lambda}(\Sigma) > 0$ appears in all of the bounds in [25, 23]). Thus, in general, the cost function $J(L)$ is not necessarily coercive in $L$, which can drastically effect the optimization landscape. For the same reason, in contrast to the LQR case [25, 23], the gradient dominant property of $J(L)$ is not clear in the filtering setup. In the next section, we show that such issues can be avoided as long as the pair $(A, H)$ is observable. Also, the learning problem posed here is distinct from its LQR counterpart (see Table 1).

## 3.2 Optimization landscape

The first result is concerned with the behaviour of the objective function at the boundary of the optimization domain. It is known [48] that the set of Schur stabilizing gains $\mathcal{S}$ is regular open, contractible, and unbounded when $m \geq 2$ and the boundary $\partial \mathcal{S}$ coincides with the set $\{L \in \mathbb{R}^{n \times m} : \rho(A - LH) = 1\}$. For simplicity of presentation, we consider a slightly stronger assumption:

**Assumption 2.** *The pair $(A, H)$ is observable, and the noise covariances $Q \succ 0$ and $R \succ 0$.*

**Lemma 1.** *The function $J(.) : \mathcal{S} \to \mathbb{R}$ is real-analytic and coercive with compact sublevel sets; i.e.,*

$$L \to \partial \mathcal{S} \text{ or } \|L\| \to \infty \text{ each implies } J(L) \to \infty,$$

*and $\mathcal{S}_\alpha := \{L \in \mathbb{R}^{n \times m} : J(L) \leq \alpha\}$ is compact and contained in $\mathcal{S}$ for any finite $\alpha > 0$.*

The next result establishes the gradient dominance property of the objective function. While this result is known in the LQR setting ([25, 23]), the extension to the estimation setup is not trivial as $H^\intercal H$, which takes the role of the covariance matrix of the initial state in LQR, may not be positive definite (instead, we only assume $(A, H)$ is observable). This, apparently minor issue, hinders establishing the gradient dominated property globally. However, we recover this property on every sublevel sets of $J$ which is sufficient for the subsequent convergence analysis.

**Lemma 2.** *Consider the constrained optimization problem* (10). *Then,*

- *The explicit formula for the gradient of $J$ is: $\nabla J(L) = 2Y_{(L)}\left(-LR + A_L X_{(L)} H^\intercal\right)$, where $Y_{(L)} = Y$ is the unique solution of $Y = A_L^\intercal Y A_L + H^\intercal H$.*

- *The global minimizer $L^* = \arg\min_{L \in \mathcal{S}} J(L)$ satisfies $L^* = AX^* H^\intercal \left(R + HX^* H^\intercal\right)^{-1}$, with $X^*$ being the unique solution of $X^* = A_{L^*} X^* A_{L^*}^\intercal + Q + L^* R (L^*)^\intercal$.*

- *The function $J(.) : \mathcal{S}_\alpha \to \mathbb{R}$, for any non-empty sublevel set $\mathcal{S}_\alpha$ for some $\alpha > 0$, satisfies the following inequalities; for all $L, L' \in \mathcal{S}_\alpha$:*

$$c_1[J(L) - J(L^*)] + c_2\|L - L^*\|_F^2 \leq \langle \nabla J(L), \nabla J(L) \rangle, \tag{11a}$$

$$c_3\|L - L^*\|_F^2 \leq J(L) - J(L^*), \tag{11b}$$

$$\|\nabla J(L) - \nabla J(L')\|_F \leq \ell \, \|L - L'\|_F, \tag{11c}$$

*for positive constants $c_1, c_2, c_3$ and $\ell$ that are only a function of $\alpha$ and independent of $L$.*

Note that the expression for the gradient is consistent with Proposition 3.8 in [23] after applying the duality relationship explained in Remark 3.

*Remark* 4. The proposition above implies that $J(.)$ has the Polyak-Łojasiewicz (PL) property (aka gradient dominance) on every $\mathcal{S}_\alpha$; i.e., for any $L \in \mathcal{S}_\alpha$ we have $J(L) - J(L^*) \leq \frac{1}{c_1(\alpha)}\langle \nabla J(L), \nabla J(L) \rangle$. The inequality (11a) is more general as it characterizes the dominance gap in terms of the iterate error from the optimality. This is useful in obtaining the iterate convergence results in the next section. Also, the Lipschitz bound resembles its "dual" counterpart in [23, Lemma 7.9], however, it is *not* implied as a simple consequence of duality because $H^\intercal H$ may not be positive definite.

## 4  SGD for Learning the Kalman Gain

In order to emphasize on the estimation time horizon $T$ for various measurement sequences, we use $\mathcal{Y}_T := \{y(t)\}_{t=0}^T$ to denote the measurement time-span. Note that, any choice of $L \in \mathcal{S}$ corresponds to a filtering strategy that outputs the following prediction,

$$\hat{y}_L(T) = HA_L^T m_0 + \sum_{t=0}^{T-1} HA_L^{T-t-1} Ly(t).$$

We denote the squared-norm of the estimation error for this filtering strategy as,

$$\varepsilon(L, \mathcal{Y}_T) := \|e_T(L)\|^2,$$

where $e_T(L) := y(T) - \hat{y}_L(T)$. We also define the *truncated* objective function as

$$J_T(L) := \mathbb{E}\left[\varepsilon(L, \mathcal{Y}_T)\right],$$

where the expectation is taken over all possible random measurement sequences, and note that, at the steady-state limit, we obtain $\lim_{T \to \infty} J_T(L) = J(L)$.

The SGD algorithm aims to solve this optimization problem by replacing the gradient, in the Gradient Descent (GD) update, with an unbiased estimate of the gradient in terms of samples from the measurement sequence. In particular, with access to an oracle that produces independent realization of the measurement sequence, say $M$ random independent measurements sequences $\{\mathcal{Y}_T^i\}_{i=1}^M$, the gradient can be approximated as follows: denote the approximated cost value

$$\widehat{J}_T(L) := \frac{1}{M}\sum_{i=1}^M \varepsilon(L, \mathcal{Y}_T^i),$$

then the approximate gradient with batch-size $M$ is $\nabla\widehat{J}_T(L) = \frac{1}{M}\sum_{i=1}^M \nabla_L\varepsilon(L, \mathcal{Y}_T^i)$. This forms an unbiased estimate of the gradient of the "truncated objective", i.e., $\mathbb{E}\left[\nabla\widehat{J}_T(L)\right] = \nabla J_T(L)$. Next, for implementation purposes, we compute the gradient estimate explicitly in terms of the measurement sequence and the filtering policy $L$.

**Lemma 3.** *Given $L \in \mathcal{S}$ and a sequence of measurements $\mathcal{Y} = \{y(t)\}_{t=0}^T$, we have,*

$$\nabla_L\varepsilon(L, \mathcal{Y}) = -2H^\intercal e_T(L)y(T-1)^\intercal$$

$$+2\sum_{t=1}^{T-1} -(A_L^\intercal)^t H^\intercal e_T(L)y(T-t-1)^\intercal + \sum_{k=1}^t (A_L^\intercal)^{t-k} H^\intercal e_T(L)y(T-t-1)^\intercal L^\intercal (A_L^\intercal)^{k-1} H^\intercal.$$

Finally, using this approximate gradient, the so-called SGD update proceeds as,

$$L_{k+1} = L_k - \eta_k \nabla_L \widehat{J}_T(L),$$

for $k \in \mathbb{Z}$, where $\eta_k > 0$ is the step-size. Numerical results of the application of this algorithm appears in Appendix G.

Table 1: Differences between SGD algorithms for optimal LQR and optimal estimation problems

| Problem | Parameters | | | Constraints | Gradient Oracle | |
|---|---|---|---|---|---|---|
| | cost value | $Q$ and $R$ | $A$ and $H$ | stability $\mathcal{S}$ | model | biased |
| LQR [25] | known | known | unknown | yes | $\mathbb{E}\left[J(L + r\Delta)\Delta\right]$ $\Delta \sim U(\mathbb{S}^{mn})$ | yes |
| Estimation (this work) | unknown | unknown | known | yes | $\mathbb{E}\left[\nabla\varepsilon(L, \mathcal{Y})\right]$ $\mathcal{Y} \sim$ output data | yes |
| Vanila SGD | * | * | * | no | $\mathbb{E}\left[\nabla\varepsilon(L, \mathcal{Y})\right]$ $\mathcal{Y} \sim$ data dist. | no |

*Remark* 5. Computing this approximate gradient only requires the knowledge of the system matrices $A$ and $H$, and does *not* require the noise covariance information $Q$ and $R$. Simulation results for the SGD algorithm are provided in the supplementary material.

Although the convergence of the SGD algorithm is expected to follow similar to the GD algorithm under the gradient dominance condition and Lipschitz property, the analysis becomes complicated due to the possibility of the iterated gain $L_k$ leaving the sub-level sets. It is expected that a convergence guarantee would hold under high-probability due to concentration of the gradient estimate around the true gradient. The complete analysis in this direction is provided in the subsequent sections.

We first provide sample complexity and convergence guarantees for SGD with a biased estimation of gradient for locally Lipschitz objective functions and in presence of stability constraint $\mathcal{S}$. Subsequently, we study the stochastic problem of estimating the gradient for the estimation problem. Distinct features of our approach as compared with similar formulations in the literature are highlighted in Table 1.

We now provide a road map to navigate the technical results that concludes with Theorem 3: Section 4.1 is concerned with the convergence analysis of the SGD algorithm under an assumption for the biased gradient oracle, summarized in Theorem 1 which concludes the linear convergence of the iterates for sufficiently small stepsize. Section 4.2 is concerned with the bias-variance error analysis of the gradient estimate, summarized in Theorem 2 providing the sufficient values for the batch-size and trajectory length that guarantees the desired bound on the gradient oracle required in Theorem 1. Finally, combining the results from Theorem 1 and Theorem 2 concludes our main result in Theorem 3.

## 4.1 SGD with biased gradient and stability constraint

First, we characterize the "robustness" of a policy at which we aim to estimate the gradient. This is formalized in the following lemma which is a consequence of [30, Lemma IV.1].

**Lemma 4.** *Consider any $L \in \mathcal{S}$ and let $Z$ be the unique solution of $Z = A_L Z A_L^\intercal + \Lambda$ for any $\Lambda \succ 0$. Then, $L + \Delta \in \mathcal{S}$ for any $\Delta \in \mathbb{R}^{n \times m}$ satisfying $0 \leq \|\Delta\|_F \leq \underline{\lambda}(\Lambda) / \left[2\,\overline{\lambda}(Z)\|H\|\right]$.*

Second, we provide a uniform lowerbound for the stepsize of gradient descent for an approximated direction "close" to the true gradient direction.

**Lemma 5** (Uniform Lower Bound on Stepsize). *Let $L_0 \in \mathcal{S}_\alpha$ for some $\alpha \geq \alpha^* := J(L^*)$, and choose any finite $\beta \geq \alpha$. Consider any direction $E$ such that $\|E - \nabla J(L_0)\|_F \leq \gamma\|\nabla J(L_0)\|_F$ for some $\gamma \in [0, 1]$, then we have $J(L_0 - \eta E) \leq \beta$ for any $\eta$ satisfying:*

$$0 \leq \eta \leq \frac{1 - \gamma}{(\gamma + 1)^2} \cdot \frac{1}{\ell(\beta)} + \frac{c_3(\alpha)}{\ell(\alpha)[\alpha - \alpha^*]} \sqrt{\frac{\beta - \alpha}{2\ell(\beta)}}.$$

*Remark* 6. Note that for the case of exact gradient direction, i.e. when $E = \nabla J(L_0)$, we have $\gamma = 0$ and choosing $\beta = \alpha$ implies the known uniform bound of $\eta \leq \frac{1}{\ell(\beta)}$ for feasible stepsizes as expected. Also, by this choice of $\beta$, this guarantees that the next iterate remains in sublevel set $\mathcal{S}_\alpha$. This lemma generalizes this uniform bound for general directions and (potentially) larger sublevel set.

The next proposition provides a decay guarantee for one iteration of gradient descent with an approximate direction which will be used later for convergence of SGD with a biased gradient estimate.

**Proposition 2** (Linear Decay in Cost Value). *Suppose $L_0 \in \mathcal{S}_\alpha$ for some $\alpha > 0$ and a direction $E \neq 0$ is given such that $\|E - \nabla J(L)\|_F \leq \gamma \|\nabla J(L)\|_F$ for some $\gamma < 1$. Let $\bar{\eta} := (1-\gamma)/(\gamma+1)^2 \ell(\alpha)$. Then, $L_1 := L_0 - \bar{\eta}E$ remains in the same sublevel set, i.e., $L_1 \in \mathcal{S}_\alpha$. Furthermore, we obtain the following linear decay of the cost value:*

$$J(L_1) - J(L^*) \leq [1 - c_1(\alpha)\bar{\eta}(1-\gamma)/2] \left[ J(L_0) - J(L^*) \right].$$

Next, we guarantee that SGD algorithm with this biased estimation of gradient obtains a linear convergence rate outside a small set $\mathcal{C}_\tau$ around optimality defined as

$$\mathcal{C}_\tau := \{ L \in \mathcal{S} \mid \|\nabla J(L)\|_F \leq s_0/\tau \},$$

for some $\tau \in (0,1)$ and arbitrarily small $s_0 > 0$. First, we assume access to the following oracle that provides a biased estimation of the true gradient.

**Assumption 3.** *Suppose, for some $\alpha > 0$, we have access to a biased estimate of the gradient $\nabla \widehat{J}(L)$ such that, there exists constants $s, s_0 > 0$ implying $\|\nabla \widehat{J}(L) - \nabla J(L)\|_F \leq s\|\nabla J(L)\|_F + s_0$ for all $L \in \mathcal{S}_\alpha \setminus \mathcal{C}_\tau$.*

**Theorem 1** (Convergence). *Suppose Assumption 3 holds with small enough $s$ and $s_0$ such that $s \leq \gamma/2$ and $\mathcal{S}_\alpha \setminus \mathcal{C}_{\gamma/2}$ is non-empty for some $\gamma \in (0,1)$. Then, SGD algorithm starting from any $L_0 \in \mathcal{S}_\alpha \setminus \mathcal{C}_{\gamma/2}$ with fixed stepsize $\bar{\eta} := \frac{(1-\gamma)}{(\gamma+1)^2 \ell(\alpha)}$ generates a sequence of policies $\{L_k\}$ that are stable (i.e. each $L_k \in \mathcal{S}_\alpha$) and cost values decay linearly before entering $\mathcal{C}_{\gamma/2}$; i.e.,*

$$J(L_k) - J(L^*) \leq [1 - c_1(\alpha)\bar{\eta}(1-\gamma)/2]^k \left[ J(L_0) - J(L^*) \right],$$

*for each $k \geq 0$ unless $L_j \in \mathcal{C}_{\gamma/2}$ for some $j \leq k$.*

*Remark* 7. By combining Theorem 1 and the PL property of the cost in (11a), we immediately obtain a sample complexity for our algorithm; e.g., choose $\gamma = 1/2$, then in order to guarantee an $\varepsilon$ error on the cost $J(L_k) - J(L^*) \leq \varepsilon$, it suffices to have a small enough bias term $s_0 \leq \frac{\sqrt{c_1(\alpha)\varepsilon}}{4}$ and variance coefficient $s \leq \frac{1}{4}$ for the oracle, and run the SGD algorithm for $k > \ln(\frac{\varepsilon}{\alpha})/\ln(1 - \frac{c_1(\alpha)}{18\ell(\alpha)}))$ steps.

### 4.2 Observation model for the estimation problem

Herein, first we show that the estimation error and its differential can be characterized as a "simple norm" of the concatenated noise (Proposition 3). This norm is induced by a metric that encapsulates the system dynamics which is explained below. Before proceeding to the results of this section, we assume that both the process and measurement noise are bounded:

**Assumption 4.** *Assume that (almost surely) $\|x_0\|, \|\xi(t)\| \leq \kappa_\xi$ and $\|\omega(t)\| \leq \kappa_\omega$ for all t. Also, for simplicity, suppose the initial state has zero mean, i.e., $m_0 = 0_n$.*

For two vectors $\vec{v}, \vec{w} \in \mathbb{R}^{(T+1)n}$, we define

$$\langle \vec{v}, \vec{w} \rangle_{\mathcal{A}_L} := \operatorname{tr}\left[ \vec{v}\vec{w}^{\mathsf{T}} \mathcal{A}_L^{\mathsf{T}} H^{\mathsf{T}} H \mathcal{A}_L \right],$$

where $\mathcal{A}_L := \begin{pmatrix} A_L^0 & A_L^1 & \ldots & A_L^T \end{pmatrix}$. Also, define $\mathcal{M}_L[E] := \begin{pmatrix} M_0[E] & M_1[E] & \cdots & M_T[E] \end{pmatrix}$ with $M_0[E] = 0$, $M_1[E] = EH$ and $M_{i+1}[E] = \sum_{k=0}^i A_L^{i-k} EH A_L^k$ for $i = 1, \cdots, T-1$.

**Proposition 3.** *The estimation error $\varepsilon(L, \mathcal{Y}_T)$ takes the following form*

$$\varepsilon(L, \mathcal{Y}_T) = \|\vec{\eta}_L\|_{\mathcal{A}_L}^2,$$

*where $\vec{\eta}_L := \vec{\xi} - (I \otimes L)\vec{\omega}$ with $\vec{\xi}^{\mathsf{T}} = \begin{pmatrix} \xi(T-1)^{\mathsf{T}} & \ldots & \xi(0)^{\mathsf{T}} & x(0)^{\mathsf{T}} \end{pmatrix}$ and $\vec{\omega}^{\mathsf{T}} = \begin{pmatrix} \omega(T-1)^{\mathsf{T}} & \ldots & \omega(0)^{\mathsf{T}} & 0_m^{\mathsf{T}} \end{pmatrix}$. Furthermore, its differential acts on small enough $E \in \mathbb{R}^{n \times m}$ as,*

$$\mathrm{d}\,\varepsilon(\cdot, \mathcal{Y}_T)\big|_L (E) = -2 \langle \vec{\eta}_L, (I \otimes E)\vec{\omega} \rangle_{\mathcal{A}_L} + \operatorname{tr}\left[ \mathcal{X}_L \mathcal{N}_L[E] \right],$$

*where $\mathcal{X}_L := \vec{\eta}_L \vec{\eta}_L^{\mathsf{T}}$ and $\mathcal{N}_L[E] := \mathcal{M}_L[E]^{\mathsf{T}} H^{\mathsf{T}} H \mathcal{A}_L + \mathcal{A}_L^{\mathsf{T}} H^{\mathsf{T}} H \mathcal{M}_L[E]$.*

Now, we want to bound the error in the estimated gradient $\nabla \widehat{J}_T(L)$ by considering the concentration error (on length $T$ trajectories) and truncation error separately as follows:

$$\|\nabla \widehat{J}_T(L) - \nabla J(L)\| \leq \|\nabla \widehat{J}_T(L) - \nabla J_T(L)\| + \|\nabla J_T(L) - \nabla J(L)\|,$$

recalling that $J_T(L) = \mathbb{E}\left[\varepsilon(L, \mathcal{Y}_T)\right]$ by definition.

Next, we aim to provide the analysis of concentration error on trajectories of length $T$ with probability bounds. However, for any pair of real $(T \times T)$-matrices $M$ and $N$, by Cauchy Schwartz inequality we obtain that $|\text{tr}[MN]| \leq \|M\|_F \|N\|_F \leq \sqrt{T}\|M\|\|N\|_F$. This bound becomes loose (in terms of dimension $T$) as the condition number of $N$ increases[2].

Nonetheless, we are able to provide concentration error bounds that "scale well with respect to the length $T$" that hinges upon the following idea: from von Neumann Trace Inequality [49, Theorem 8.7.6] one obtains,

$$|\text{tr}[MN]| \leq \sum_{i=1}^{T} \sigma_i(M)\sigma_i(N) \leq \|M\|\|N\|_*, \tag{12}$$

where $\|N\|_* := \text{tr}\left[\sqrt{N^\intercal N}\right] = \sum_i \sigma_i(N)$ is the *nuclear norm* with $\sigma_i(N)$ denoting the $i$-th largest singular value of $N$. Additionally, the same inequality holds for non-square matrices of appropriate dimension which is tight in terms of dimension.

**Proposition 4** (Concentration Error Bound). *Consider $M$ independent length $T$ trajectories $\{\mathcal{Y}_T^i\}_{i=1}^{M}$ and suppose Assumption 4 holds. Let $\nabla \widehat{J}_T(L) := \frac{1}{M} \sum_{i=1}^{M} \nabla \varepsilon(L, \mathcal{Y}_T^i)$, then for any $s > 0$,*

$$\mathbb{P}\left[\|\nabla \widehat{J}_T(L) - \nabla J_T(L)\| \geq s\right] \leq 2n \exp\left[\frac{-Ms^2/2}{\nu_L^2 + 2\nu_L s/3}\right],$$

*where $\nu_L := 4\kappa_L^2 C_L^3 \|H\|^2 \|H\|_* / [1 - \sqrt{\rho(A_L)}]^3$ with $\kappa_L = \kappa_\xi + \|L\|\kappa_\omega$.*

We can also show how the truncation error decays linearly as $T$ grows, with constants that are independent of the system dimension $n$:

**Proposition 5** (Truncation Error Bound). *Under Assumption 4, the truncation error is bounded:*

$$\|\nabla J(L) - \nabla J_T(L)\| \leq \bar{\gamma}_L \sqrt{\rho(A_L)}^{T+1},$$

*where $\bar{\gamma}_L := 10\kappa_L^4 C_L^6 \|H\|^2 \|H\|_* / [1 - \rho(A_L)]^2$.*

*Remark* 8. Notice how the trajectory length $T$ determines the bias in the estimated gradient. However, the concentration error bound is independent of $T$ and only depends on the noise bounds proportionate to $\kappa_\xi^4, \kappa_\omega^4$ and the stability margin of $A_L$ proportionate to $C_L^6 / (1 - \sqrt{\rho(A_L)})^6$.

Finally, by combining the truncation bound in Proposition 5 with concentration bounds in Proposition 4 we can provide probabilistic bounds on the "estimated cost" $\widehat{J}_T(L)$ and the "estimated gradient" $\nabla \widehat{J}_T(L)$. Its precise statement is deferred to the supplementary materials (Theorem 5).

## 4.3 Sample complexity of SGD for Kalman Gain

Note that the open-loop system may be unstable. Often in learning literature, it is assumed that the closed-loop system can be contractible ( i.e., the spectral norm $\|A_L\| < 1$) which is quite convenient for analysis, however, it is not a reasonable system theoretic assumption. Herein, we emphasize that we only require the close-loop system to be Schur stable, meaning that $\rho(A_L) < 1$; yet, it is very well possible that the system is not contractible. Handling systems that are merely stable requires more involved system theoretic tools that are established in the following lemma.

**Lemma 6** (Uniform Bounds for Stable Systems). *Suppose $L \in \mathcal{S}$, then there exit a constants $C_L > 0$ such that*

$$\|A_L^k\| \leq C_L \sqrt{\rho(A_L)}^{k+1}, \quad \forall k \geq 0.$$

*Furthermore, consider $\mathcal{S}_\alpha$ for some $\alpha > 0$, then there exist constants $D_\alpha > 0$, $C_\alpha > 0$ and $\rho_\alpha \in (0, 1)$ such that $\|L\| \leq D_\alpha$, $C_L \leq C_\alpha$, and $\rho(A_L) \leq \rho_\alpha$ for all $L \in \mathcal{S}_\alpha$.*

---

[2]The reason is that the first equality is sharp whenever $M$ is in the "direction" of $N$, while the second inequality is sharp whenever condition number of $M$ is close to one.

The following result provides sample complexity bounds for this stochastic oracle to provide a biased estimation of the gradient that satisfies our oracle model of SGD analysis in Assumption 3.

**Theorem 2.** *Under the premise of Proposition 4, consider $\mathcal{S}_\alpha$ for some $\alpha > 0$ and choose any $s, s_0 > 0$ and $\tau \in (0, 1)$. Suppose the trajectory length $T \geq \ln\left(\bar{\gamma}_\alpha \sqrt{\min(n, m)}/s_0\right) \Big/ \ln\left(1/\sqrt{\rho_\alpha}\right)$ and the batch size $M \geq 4\nu_\alpha^2 \min(n, m) \ln(2n/\delta)/(s\, s_0)^2$, where $\bar{\gamma}_\alpha := 10(\kappa_\xi + D_\alpha \kappa_\omega)^4 C_\alpha^6 \|H\|^2 \|H\|_*$ and $\nu_\alpha := 5C_\alpha^3 \|H\|^2 \|H\|_* (\kappa_\xi + D_\alpha \kappa_\omega)^2 / [1 - \sqrt{\rho_\alpha}]^3$. Then, with probability no less than $1 - \delta$, Assumption 3 holds.*

*Proof.* First, note that for any $L \in \mathcal{S}_\alpha$ by Lemma 6 we have that $\bar{\gamma}_L \leq \bar{\gamma}_\alpha$ and $\nu_L \leq \nu_\alpha$. Then, note that the lower bound on $T$ implies that $\bar{\gamma}_\alpha \sqrt{\min(n, m)}\sqrt{\rho_\alpha}^{T+1} \leq s_0$. The claim then follows by applying Theorem 5 and noting that for any $L \notin \mathcal{C}_\tau$ the gradient is lowerbound as $\|\nabla J(L)\| > s_0/\tau$. $\qquad\square$

**Theorem 3.** *Consider the linear system (1) under Assumptions 2 and 4. Suppose the SGD algorithm is implemented with initial stabilizing gain $L_0$, the step-size $\bar{\eta} := \frac{2}{9\ell(J(L_0))}$, for $k$ iterations, a batch-size of $M$, and data-length $T$. Then, $\forall \varepsilon > 0$ and with probability larger than $1 - \delta$, $J(L_k) - J(L^*) \leq \varepsilon$ if*

$$T \geq O(\ln(\frac{1}{\varepsilon})), \quad M \geq O(\frac{1}{\varepsilon}\ln(\frac{1}{\delta})\ln(\ln(\frac{1}{\varepsilon}))), \quad and \quad k \geq O(\ln(\frac{1}{\varepsilon})). \tag{13}$$

*Proof of Theorem 3.* Recall that according to Remark 7, in order to obtain $\varepsilon$ error on the cost value we require $k \geq O(\ln(1/\varepsilon))$ of SGD algorithm using gradient estimates with small enough bias term $s_0 \leq \frac{\sqrt{c_1(\alpha)\varepsilon}}{4}$ and variance coefficient $s < \frac{1}{4}$. Now, we can guarantee Assumption 3 holds for such $s_0$ and $s$ (with probability at least $1 - \delta$) by invoking Theorem 2. In fact, it suffices to ensure that the length of data trajectories $T$ and the batch size $M$ are large enough; specifically, $T \geq O(\ln(1/s_0)) = O(\ln(1/\varepsilon))$ and by using union bound for bounding the failure probability $M \geq O(\ln(1/\delta)\ln(k)/s_0^2) = O(\ln(1/\delta)\ln(\ln(1/\varepsilon))/\varepsilon)$. $\qquad\square$

## 5 Conclusions, Broader Impact, and Limitations

In this work, we considered the problem of learning the optimal (steady-state) Kalman gain for linear systems with unknown process and measurement noise covariances. Our approach builds on the duality between optimal control and estimation, resulting in a direct stochastic Policy Optimization (PO) algorithm for learning the optimal filter gain. We also provided convergence guarantees and finite sample complexity with bias and variance error bounds that scale well with problem parameters. In particular, the variance is independent of the length of data trajectories and scales logarithmically with problem dimension, and bias term decreases exponentially with the length.

This work contributes a generic optimization algorithm and introduces a filtering strategy for estimating dynamical system states. While theoretical, it raises privacy concerns similar to the model-based Kalman filter. Limitations include the need for prior knowledge of system parameters; nonetheless, parameter uncertainties can be treated practically as process and measurement noise. Finally, sample complexities depend on the stability margin $1 - \sqrt{\rho(A_L)}$, inherent to the system generating the data.

Finally, a direction for future research is to study how to adapt the proposed algorithm and error analysis for the setting when a single long trajectory is available as opposed to several independent trajectories of finite length. Another interesting direction is to carry out a robustness analysis, similar to its LQR dual counterpart, to study the effect of the error in system parameters on the policy learning accuracy. Of course, the ultimate research goal is to use the recently introduced duality in nonlinear filtering [47] as a bridge to bring tools from learning to nonlinear filtering.

## Acknowledgments and Disclosure of Funding

The research of A. Taghvaei has been supported by NSF grant EPCN-2318977; S. Talebi and M. Mesbahi have been supported by NSF grant ECCS-2149470 and AFOSR grant FA9550-20-1-0053. The authors are also grateful for constructive feedback from the reviewers.

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

# Contents

## Appendix A   Discussion of additional related works

After the initial submission of this manuscript, we encountered the following related works, and we will now briefly discuss their connections and differences in relation to our work: [36] establishes an end-to-end sample complexity bound on learning a robust LQG controller (which is different than our filtering problem). As the system parameters are also unknown, this work only considers the *open-loop stable* systems. Additionally, it establishes a nice trade-off between optimality and robustness characterized by a suboptimality performance bound as a result of the robust LQG design. While our filtering design is based on the knowledge of system parameters, we do not consider the open-loop stability assumption; and the robust synthesis is included as one of the main future directions of this work. Furthermore, the complexity bounds in [36] depends on the length of trajectory, whereas ours does not. Also, their suboptimality bound scale as $O(1/\sqrt{N})$ in the number of trajectories, whereas ours scales as $O(1/N)$. Next, [37] similarly considers the problem of learning the steady-state Kalman gain but in a different setup: The model is assumed to be completely unknown. However, the algorithm requires access to a simulator that provides noisy measurement of the MSE $\mathbb{E}[\|X(t) - \hat{X}(t)\|^2]$ which requires generation of ground-truth state trajectories $X(t)$ (see Assumption 3.2 therein). The proposed approaches are different: zeroth-order global optimization (theirs) vs first-order stochastic gradient descent (ours). The analysis reports a $O(\varepsilon^{-2})$ sample complexity on optimal policy error which aligns with our result. However, it is difficult to provide more detailed comparison as explicit dependence of the error terms on problem dimension is not provided. Finally, [38] considers the problem of simultaneously learning the optimal Kalman filter and linear feedback control policy in the LQG setup. Their approach involves solving Semidefinite Programmings (SDP) using input-state-output trajectories. Their result, for the the case when trajectories involve noise, relies on the assumption that the magnitude of the noise can be made arbitrary small. This is in contrast to our setup where we only assume a bound on the noise level and do not require access to state trajectories.

## Appendix B   Summary of Assumptions

Herein, we provide a summary of all assumptions considered in our work and their relation to our results.

Note that if the given pair $(A, H)$ of system parameters is not detectable, then the estimation problem is not well-posed, simply because there may not exist any stabilizing policy $L_0$. Therefore, we consider the minimum required assumption for well-posedness of the problem as follows:

**Assumption 1.** *The pair $(A, H)$ is detectable, and the pair $(A, Q^{\frac{1}{2}})$ is stabilizable, where $Q^{\frac{1}{2}}$ is the unique positive semidefinite square root of $Q$.*

It is indeed possible to work with the weaker conditions compared with Assumption 1 and obtain similar results presented here—by incorporating more system theoretic tools. However, in order to improve the clarity of the learning problem with less system theoretic technicalities, we work with the following stronger assumption in lieu of Assumption 1.

**Assumption 2.** *The pair $(A, H)$ is observable, and the noise covariances $Q \succ 0$ and $R \succ 0$.*

It is clear that Assumption 2 implies Assumption 1.

Next, in order to provide an independent analysis of the SGD algorithm for locally Lipschitz functions in presence of gradient bias, we consider the following assumption on the gradient oracle.

**Assumption 3.** *Suppose, for some $\alpha > 0$, we have access to a biased estimate of the gradient $\nabla \widehat{J}(L)$ such that, there exists constants $s, s_0 > 0$ implying $\|\nabla \widehat{J}(L) - \nabla J(L)\|_F \leq s \|\nabla J(L)\|_F + s_0$ for all $L \in \mathcal{S}_\alpha \setminus \mathcal{C}_\tau$.*

Next, in order to utilize the complexity analysis of SGD algorithm provided in Theorem 1, we need to build an oracle that satisfies Assumption 3. This is oracle is built in Section 4.2, building on the following assumptions on the process and observation noise.

**Assumption 4.** *Assume that (almost surely) $\|x_0\|, \|\xi(t)\| \leq \kappa_\xi$ and $\|\omega(t)\| \leq \kappa_\omega$ for all t. Also, for simplicity, suppose the initial state has zero mean, i.e., $m_0 = 0_n$.*

Then, based on this assumption, Theorem 2 provides lowerbounds on the trajectory length and the batch-size in order for the oracle to satisfy Assumption 3. Finally, the combined version of our results is presented in Theorem 3.

## Appendix C   Proof of the duality relationship: Proposition 1

1. By pairing the original state dynamics (1) and its dual (6):
$$z(t+1)^\intercal x(t+1) - z(t)^\intercal x(t) = z(t+1)^\intercal \xi(t) + u(t+1)^\intercal H x(t).$$
Summing this relationship from $t = 0$ to $t = T - 1$ yields,
$$z(T)^\intercal x(T) = z(0)^\intercal x(0) + \sum_{t=0}^{T-1} z(t+1)^\intercal \xi(t) + u(t+1)^\intercal H x(t).$$
Upon subtracting the estimate $a^\intercal \hat{x}_\mathcal{L}(T)$, using the adjoint relationship
$$\sum_{t=0}^{T-1} u(t+1)^\intercal y(t) = a^\intercal \hat{x}_\mathcal{L}(T). \tag{14}$$
and $z(T) = a$, lead to
$$a^\intercal x(T) - a^\intercal \hat{x}_\mathcal{L}(T) = z(0)^\intercal x(0) + \sum_{t=0}^{T-1} z(t+1)^\intercal \xi(t) - u(t+1)^\intercal w(t).$$
Squaring both sides and taking the expectation concludes the duality result in (8).

2. Consider the adjoint system (6) with the linear feedback law $u(t) = L^\intercal z(t)$. Then,
$$z(t) = (A_L^\intercal)^{T-t} a, \quad \text{for} \quad t = 0, 1, \ldots, T. \tag{15}$$
Therefore, as a function of $a$, $u(t) = L^\intercal (A_L^\intercal)^{T-t} a$. These relationships are used to identify the control policy
$$\mathcal{L}^\dagger(a) = (u(1), \ldots, u(T)) = (L^\intercal (A_L^\intercal)^{T-1} a, \ldots, L^\intercal a).$$
This control policy corresponds to an estimation policy by the adjoint relationship (14):
$$a^\intercal \hat{x}_\mathcal{L}(T) = \sum_{t=0}^{T-1} a^\intercal A_L^{T-t-1} L y(t), \quad \forall a \in \mathbb{R}^n.$$
This relationship holds for all $a \in \mathbb{R}^n$. Therefore,
$$\hat{x}_\mathcal{L}(T) = \sum_{t=0}^{T-1} A_L^{T-t-1} L y(t),$$
which coincides with the Kalman filter estimate with constant gain $L$ given by the formula (4) (with $m_0 = 0$). Therefore, the adjoint relationship (14) relates the control policy with constant gain $L^\intercal$ to the Kalman filter with constant gain $L$. The result (9) follows from the identity
$$\mathbb{E}\left[\|y(T) - \hat{y}_L(T)\|^2\right] = \mathbb{E}\left[\|H x(T) - H \hat{x}_L(T)\|^2 + \|w(T)\|^2\right]$$
$$= \sum_{i=1}^m \mathbb{E}\left[|H_i^\intercal x(T) - H_i^\intercal \hat{x}_L(T)|^2\right] + \text{tr}[R],$$
and the application of the first result (9) with $a = H_i$.

## Appendix D  Proofs for the results for the analysis of the optimization landscape

### D.1  Preliminary lemma

The following lemmas are a direct consequence of duality and useful for our subsequent analysis.

**Lemma 7.** *The set of Schur stabilizing gains $\mathcal{S}$ is regular open, contractible, and unbounded when $m \geq 2$ and the boundary $\partial \mathcal{S}$ coincides with the set $\{L \in \mathbb{R}^{n \times m} : \rho(A - LH) = 1\}$. Furthermore, $J(.)$ is real analytic on $\mathcal{S}$.*

*Proof.* Consider the duality described in Remark 3. The proof then follows identical to [23, Lemmas 3.5 and 3.6] by noting that the spectrum of a matrix is identical to the spectrum of its transpose. $\square$

Next, we present the proof of Lemma 1 that provides sufficient conditions to recover the coercive property of $J(.)$ which resembles Lemma 3.7 in [23] (but extended for the time-varying parameters). The proof is removed on the account of space and appears in the extended version of this work.

*Remark* 9. This approach recovers the claimed coercivity also in the control setting with weaker assumptions. In particular, using this result, one can replace the positive definite condition on the covariance of the initial condition in [23], i.e., $\Sigma \succ 0$, with just the controllability of $(A, \Sigma^{1/2})$.

### D.2  Proof of Lemma 1

*Proof.* Consider any $L \in \mathcal{S}$ and note that the right eigenvectors of $A$ and $A_L$ that are annihilated by $H$ are identical. Thus, by Popov-Belevitch-Hautus (PBH) test, observability of $(A, H)$ is equivalent to observability of $(A_L, H)$. Therefore, there exists a positive integer $n_0 \leq n$ such that

$$H_{n_0}^{\intercal}(L) := \begin{pmatrix} H^{\intercal} & A_L^{\intercal} H^{\intercal} & \cdots & (A_L^{\intercal})^{n_0-1} H^{\intercal} \end{pmatrix}$$

is full-rank, implying that $H_{n_0}^{\intercal}(L)H_{n_0}(L)$ is positive definite. Now, recall that for any such stabilizing gain $L$, we compute

$$J(L) = \operatorname{tr}\left[\sum_{t=0}^{\infty} (A_L)^t (Q + LRL^{\intercal})(A_L^{\intercal})^t H^{\intercal} H\right]$$

$$= \operatorname{tr}\left[\sum_{t=0}^{\infty} \sum_{k=0}^{n_0-1} (A_L)^{n_0 t + k}(Q + LRL^{\intercal})(A_L^{\intercal})^{n_0 t + k} H^{\intercal} H\right]$$

$$= \operatorname{tr}\left[\sum_{t=0}^{\infty} (A_L)^{n_0 t}(Q + LRL^{\intercal})(A_L^{\intercal})^{n_0 t} H_{n_0}^{\intercal}(L)H_{n_0}(L)\right]$$

$$=: \operatorname{tr}\left[X_{n_0}(L)H_{n_0}^{\intercal}(L)H_{n_0}(L)\right],$$

where we used the cyclic property of trace and the inequality follows because for any PSD matrices $P_1, P_2 \succeq 0$ we have

$$\operatorname{tr}[P_1 P_2] = \operatorname{tr}\left[P_2^{\frac{1}{2}} P_1 P_2^{\frac{1}{2}}\right] \geq 0. \tag{16}$$

Also, $X_{n_0}(L)$ is well defined because $A_L$ is Schur stable if and only if $(A_L)^{n_0}$ is. Moreover, $X_{n_0}(L)$ coincides with the unique solution to the following Lyapunov equation

$$X_{n_0}(L) = (A_L)^{n_0} X_{n_0}(L)(A_L^{\intercal})^{n_0} + Q + LRL^{\intercal}.$$

Next, as $R \succeq 0$,

$$J(L) \geq \underline{\lambda}(H_{n_0}^{\intercal}(L)H_{n_0}(L))\operatorname{tr}[X_{n_0}(L)]$$

$$\geq \underline{\lambda}(H_{n_0}^{\intercal}(L)H_{n_0}(L))\operatorname{tr}\left[\sum_{t=0}^{\infty}(A_L)^{n_0 t} Q (A_L^{\intercal})^{n_0 t}\right]$$

$$\geq \underline{\lambda}(H_{n_0}^{\intercal}(L)H_{n_0}(L))\,\underline{\lambda}(Q)\sum_{t=0}^{\infty}\operatorname{tr}\left[(A_L^{\intercal})^{n_0 t}(A_L)^{n_0 t}\right]$$

$$\geq \underline{\lambda}(H_{n_0}^{\intercal}(L)H_{n_0}(L))\,\underline{\lambda}(Q)\sum_{t=0}^{\infty}\rho(A_L)^{2n_0 t}, \tag{17}$$

where the last inequality follows by the fact that

$$\operatorname{tr}\left[(A_L^{\intercal})^{n_0 t}(A_L)^{n_0 t}\right] \;=\; \|(A_L)^{n_0 t}\|_F^2 \;\geq\; \|(A_L)^{n_0 t}\|_{\mathrm{op}}^2 \;\geq\; \rho((A_L)^{n_0 t})^2 \;=\; \rho(A_L)^{2 n_0 t},$$

with $\|\cdot\|_{\mathrm{op}}$ denoting the operator norm induced by 2-norm. Now, by Lemma 7 and continuity of the spectral radius, as $L_k \to \partial S$ we observe that $\rho(A_{L_k}) \to 1$. But then, the obtained lowerbound implies that $J(L_k) \to \infty$. On the other hand, as $Q \succ 0$, $R \succ 0$ are both time-independent, by using a similar technique we also provide the following lowerbound

$$
\begin{aligned}
J(L) \geq & \operatorname{tr}\left[(Q + LRL^{\intercal})\sum_{t=0}^{\infty}(A_L^{\intercal})^{n_0 t} H_{n_0}^{\intercal}(L) H_{n_0}(L)(A_L)^{n_0 t}\right] \\
\geq & \underline{\lambda}(H_{n_0}^{\intercal}(L) H_{n_0}(L))\operatorname{tr}\left[Q + LRL^{\intercal}\right] \\
\geq & \underline{\lambda}(H_{n_0}^{\intercal}(L) H_{n_0}(L))\operatorname{tr}\left[RLL^{\intercal}\right] \\
\geq & \underline{\lambda}(H_{n_0}^{\intercal}(L) H_{n_0}(L))\,\underline{\lambda}(R)\|L\|_F^2,
\end{aligned}
$$

where $\|\cdot\|_F$ denotes the Frobenius norm. Therefore, by equivalency of norms on finite dimensional spaces, $\|L_k\| \to \infty$ implies that $J(L_k) \to \infty$ which concludes that $J(.)$ is coercive on $S$. Finally, note that for any $L \notin S$, by (17) we can argue that $J(L) = \infty$, therefore the sublevel sets $S_\alpha \subset S$ whenever $\alpha$ is finite. The compactness of $S_\alpha$ is then a direct consequence of the coercive property and continuity of $J(.)$ (Lemma 7). □

### D.3   Derivation of the gradient formula in Lemma 2

Next, we aim to compute the gradient of $J$ in a more general format. We do the derivation for time-varying $R$ and $Q$ with specialization to time-invariant setting at the end. For any admissible $\Delta$, we have

$$
\begin{aligned}
X_\infty(L + \Delta) - X_\infty(L) = &\sum_{t=1}^{\infty}(A_L)^t (Q_t + LR_t L^{\intercal}) (\star)^{\intercal} + (\star)(Q_t + LR_t L^{\intercal})(A_L^{\intercal})^t \\
& - \sum_{t=0}^{\infty}(A_L)^t (\Delta R_t L^{\intercal} + LR_t \Delta^{\intercal})(A_L^{\intercal})^t + o(\|\Delta\|),
\end{aligned}
$$

where the $\star$ is hiding the following term

$$\sum_{k=1}^{t}(A_L)^{t-k}\Delta H (A_L)^{k-1}.$$

Therefore, by linearity and cyclic permutation property of trace, we get that

$$
\begin{aligned}
J(L + \Delta) - J(L) = & \operatorname{tr}\left[\Delta H \sum_{t=1}^{\infty}\sum_{k=1}^{t} 2 (A_L)^{k-1}(Q_t + LR_t L^{\intercal})(A_L^{\intercal})^t H^{\intercal} H (A_L)^{t-k}\right] \\
& - \operatorname{tr}\left[\Delta \sum_{t=0}^{\infty} 2R_t L^{\intercal}(A_L^{\intercal})^t H^{\intercal} H (A_L)^t\right] + o(\|\Delta\|).
\end{aligned}
$$

Finally, by considering the Euclidean metric on real matrices induced by the inner product $\langle Q, P\rangle = \operatorname{tr}\left[Q^{\intercal}P\right]$, we obtain the gradient of $J$ as follows

$$
\begin{aligned}
\nabla J(L) = & -2\sum_{t=0}^{\infty}(A_L^{\intercal})^t H^{\intercal} H (A_L)^t LR_t \\
& +2\sum_{t=1}^{\infty}\sum_{k=1}^{t}(A_L^{\intercal})^{t-k} H^{\intercal} H (A_L)^t (Q_t + LR_t L^{\intercal})(A_L^{\intercal})^{k-1} H^{\intercal},
\end{aligned}
$$

whenever the series are convergent! And, by switching the order of the sums it simplifies to

$$
\nabla J(L) = -2 \sum_{t=0}^{\infty} \left( A_L^\intercal \right)^t H^\intercal H \left( A_L \right)^t L R_t
$$

$$
+ 2 \sum_{k=1}^{\infty} \sum_{t=k}^{\infty} \left[ \left( A_L^\intercal \right)^{t-k} H^\intercal H \left( A_L \right)^{t-k} \right] A_L \left[ \left( A_L \right)^{k-1} \left( Q_t + L R_t L^\intercal \right) \left( A_L^\intercal \right)^{k-1} \right] H^\intercal.
$$

$$
= -2 \sum_{t=0}^{\infty} \left( A_L^\intercal \right)^t H^\intercal H \left( A_L \right)^t L R_t + 2 \sum_{t=0}^{\infty} \left[ \left( A_L^\intercal \right)^t H^\intercal H \left( A_L \right)^t \right]
$$

$$
\cdot A_L \left[ \sum_{k=0}^{\infty} \left( A_L \right)^k \left( Q_{t+k+1} + L R_{t+k+1} L^\intercal \right) \left( A_L^\intercal \right)^k \right] H^\intercal.
$$

For the case of time-independent $Q$ and $R$, this reduces to

$$
\nabla J(L) = -2 Y_{(L)} L R + 2 Y_{(L)} A_L \left[ \sum_{k=0}^{\infty} \left( A_L \right)^k \left( Q + L R L^\intercal \right) \left( A_L^\intercal \right)^k \right] H^\intercal
$$

$$
= 2 Y_{(L)} \left[ -L R + A_L X_{(L)} H^\intercal \right].
$$

where $Y_{(L)} = Y$ is the unique solution of

$$
Y = A_L^\intercal Y A_L + H^\intercal H.
$$

## D.4  Proof of existence of global minimizer in Lemma 2

The domain $\mathcal{S}$ is non-empty whenever $(A, H)$ is observable. Thus, by continuity of $L \to J(L)$, there exists some finite $\alpha > 0$ such that the sublevel set $\mathcal{S}_\alpha$ is non-empty and compact. Therefore, the minimizer is an interior point and thus must satisfy the first-order optimality condition $\nabla J(L^*) = 0$. Therefore, by coercivity, it is stabilizing and unique which satisfies

$$
L^* = A X^* H^\intercal \left( R + H X^* H^\intercal \right)^{-1},
$$

with $X^*$ being the unique solution of

$$
X^* = A_{L^*} X^* A_{L^*}^\intercal + Q + L^* R (L^*)^\intercal. \tag{18}
$$

As expected, the global minimizer $L^*$ is equal to the steady-state Kalman gain, but explicitly dependent on the noise covariances $Q$ and $R$.

## D.5  Proof of gradient dominance property in Lemma 2

Note that $X = X_{(L)}$ satisfies

$$
X = A_L X A_L^\intercal + Q + L R L^\intercal. \tag{19}
$$

Then, by combining (18) and (19), and some algebraic manipulation, we recover part of the gradient information, i.e., $(-LR + A_L X H^\intercal)$, in the gap of cost matrices by arriving at the following identity

$$
\begin{aligned}
& X - X^* - A_{L^*} (X - X^*) A_{L^*}^\intercal \\
&= (LR - A_L X H^\intercal)(L - L^*)^\intercal + (L - L^*)(R L^\intercal - H X A_L^\intercal) \\
&\quad - (L - L^*) R (L - L^*)^\intercal - (L - L^*) H X H^\intercal (L - L^*)^\intercal \\
&\preceq \frac{1}{a} (LR - A_L X H^\intercal)(R L^\intercal - H X A_L^\intercal) + a(L - L^*)(L - L^*)^\intercal \\
&\quad - (L - L^*)(R + H X H^\intercal)(L - L^*)^\intercal
\end{aligned} \tag{20}
$$

where the upperbound is valid for any choice of $a > 0$. Now, as $R \succ 0$, we choose $a = \underline{\lambda}(R)/2$. As $X \succeq 0$, it further upperbounds

$$
\begin{aligned}
& X - X^* - A_{L^*} (X - X^*) A_{L^*}^\intercal \\
&\preceq \frac{2}{\underline{\lambda}(R)} (-LR + A_L X H^\intercal)(-R L^\intercal + H X A_L^\intercal) \\
&\quad - \frac{\underline{\lambda}(R)}{2} (L - L^*)(L - L^*)^\intercal.
\end{aligned}
$$

Now, let $\tilde{X}$ and $\widehat{X}$ be, respectively, the unique solution of the following Lyapunov equations

$$\tilde{X} = A_{L^*}\tilde{X}A_{L^*}^\mathsf{T} + (-LR + A_L X H^\mathsf{T})(-RL^\mathsf{T} + HXA_L^\mathsf{T}),$$
$$\widehat{X} = A_{L^*}\widehat{X}A_{L^*}^\mathsf{T} + (L - L^*)(L - L^*)^\mathsf{T}.$$

Then by comparison, we conclude that

$$X - X^* \preceq \frac{2}{\underline{\lambda}(R)}\tilde{X} - \frac{\underline{\lambda}(R)}{2}\widehat{X}.$$

Recall that by the fact in (16),

$$J(L) - J(L^*) = \operatorname{tr}\left[(X - X^*)H^\mathsf{T}H\right] \leq \frac{2}{\underline{\lambda}(R)}\operatorname{tr}\left[\tilde{X}H^\mathsf{T}H\right] - \frac{\underline{\lambda}(R)}{2}\operatorname{tr}\left[\widehat{X}H^\mathsf{T}H\right]. \quad (21)$$

Let $Y^* \succ 0$ be the unique solution of

$$Y^* = A_{L^*}^\mathsf{T}Y^*A_{L^*} + H^\mathsf{T}H,$$

then, by cyclic permutation property

$$\operatorname{tr}\left[\tilde{X}H^\mathsf{T}H\right] = \operatorname{tr}\left[(-LR + A_L X H^\mathsf{T})(-RL^\mathsf{T} + HXA_L^\mathsf{T})Y^*\right]$$

$$\leq \frac{\overline{\lambda}(Y^*)}{\underline{\lambda}^2(Y_{(L)})}\operatorname{tr}\left[Y_{(L)}(-LR + A_L X H^\mathsf{T})(-RL^\mathsf{T} + HXA_L^\mathsf{T})Y_{(L)}\right]$$

$$= \frac{\overline{\lambda}(Y^*)}{4\,\underline{\lambda}^2(Y_{(L)})}\langle\nabla J(L), \nabla J(L)\rangle \quad (22)$$

where the last equality follows by the obtained formula for the gradient $\nabla J(L)$. Similarly, we obtain that

$$\operatorname{tr}\left[\widehat{X}H^\mathsf{T}H\right] = \operatorname{tr}\left[(L - L^*)(L - L^*)^\mathsf{T}Y^*\right] \geq \underline{\lambda}(Y^*)\|L - L^*\|_F^2. \quad (23)$$

Notice that the mapping $L \to Y_{(L)}$ is continuous on $\mathcal{S} \supset \mathcal{S}_\alpha$, and also by observability of $(A, H)$, $Y_{(L)} \succ 0$ for any $L \in \mathcal{S}$. To see this, let $H_{n_0}(L) \succ 0$ be as defined in Lemma 1. Then,

$$Y_{(L)} = \sum_{t=0}^{\infty}(A_L^\mathsf{T})^t(H^\mathsf{T}H)(A_L)^t$$

$$= \sum_{t=0}^{\infty}\sum_{k=0}^{n_0-1}(A_L^\mathsf{T})^{n_0 t + k}(H^\mathsf{T}H)(A_L)^{n_0 t + k}$$

$$= \sum_{t=0}^{\infty}(A_L^\mathsf{T})^{n_0 t}H_{n_0}^\mathsf{T}(L)H_{n_0}(L)(A_L)^{n_0 t}$$

$$\succeq H_{n_0}^\mathsf{T}(L)H_{n_0}(L) \succ 0.$$

Now, by Lemma 1, $\mathcal{S}_\alpha$ is compact and therefore we claim that the following infimum is attained with some positive value $\kappa_\alpha$:

$$\inf_{L \in \mathcal{S}_\alpha}\underline{\lambda}(Y_{(L)}) =: \kappa_\alpha > 0. \quad (24)$$

Finally, the first claimed inequality follows by combining the inequalities (21), (22) and (23), with the following choice of

$$c_1(\alpha) := \frac{2\,\underline{\lambda}(R)}{\overline{\lambda}(Y^*)}\kappa_\alpha^2, \quad \text{and} \quad c_2(\alpha) := \frac{\underline{\lambda}(Y^*)\,\underline{\lambda}(R)^2}{\overline{\lambda}(Y^*)}\kappa_\alpha^2.$$

For the second claimed inequality, one arrive at the following identity by similar computation to (20):

$$X - X^* - A_L(X - X^*)A_L^\mathsf{T}$$
$$= (L^*R - A_{L^*}X^*H^\mathsf{T})(L - L^*)^\mathsf{T} + (L - L^*)(RL^{*\mathsf{T}} - HX^*A_{L^*}^\mathsf{T})$$
$$\quad + (L - L^*)R(L - L^*)^\mathsf{T} + (L - L^*)HX^*H^\mathsf{T}(L - L^*)^\mathsf{T}$$
$$= (L - L^*)(R + HX^*H^\mathsf{T})(L - L^*)^\mathsf{T}$$

where the second equality follows because $Y_{(L)} \succ 0$ and thus

$$L^* R - A_{L^*} X^* H^\intercal = -Y_{(L)}^{-1} \nabla J(L^*) = 0.$$

Recall that

$$J(L) - J(L^*) = \mathrm{tr}\left[(X - X^*) H^\intercal H\right],$$

then by the equality in (20) and cyclic property of trace we obtain

$$J(L) - J(L^*) = \mathrm{tr}\left[Z Y_{(L)}\right],$$

where

$$\begin{aligned}
Z :=& (L - L^*)(R + H X^* H^\intercal)(L - L^*)^\intercal \\
\succeq& \underline{\lambda}(R)(L - L^*)(L - L^*)^\intercal.
\end{aligned}$$

Therefore, for any $L \in \mathcal{S}_\alpha$, we have

$$J(L) - J(L^*) \geq \underline{\lambda}(Y_{(L)}) \mathrm{tr}\left[Z\right] \geq \underline{\lambda}(R) \kappa_\alpha \| L - L^* \|_F^2,$$

and thus, we complete the proof of first part by the following choice of

$$c_3(\alpha) = \underline{\lambda}(R) \kappa_\alpha.$$

### D.6 Proof of Lipschitz property in Lemma 2

Next, we provide the proof of locally Lipschitz property. Notice that the mappings $L \to X_{(L)}$, $L \to Y_{(L)}$ and $L \to A_L$ are all real-analytic on the open set $\mathcal{S} \supset \mathcal{S}_\alpha$, and thus so is the mapping $L \to \nabla J(L) = 2 Y_{(L)} \left[-LR + A_L X_{(L)} H^\intercal\right]$. Also, by Lemma 1, $\mathcal{S}_\alpha$ is compact and therefore the mapping $L \to \nabla J(L)$ is $\ell$-Lipschitz continuous on $\mathcal{S}_\alpha$ for some $\ell = \ell(\alpha) > 0$. In the rest of the proof, we attempt to characterize $\ell(\alpha)$ in terms of the problem parameters. By direct computation we obtain

$$\begin{aligned}
\nabla J(L_1) - \nabla J(L_2) =& (2 Y_{(L_1)} - 2 Y_{(L_2)}) \left[-L_1 R + A_{L_1} X_{(L_1)} H^\intercal\right] \\
&+ 2 Y_{(L_2)} \left(\left[-L_1 R + A_{L_1} X_{(L_1)} H^\intercal\right] - \left[-L_2 R + A_{L_2} X_{(L_2)} H^\intercal\right]\right) \\
=& 2(Y_{(L_1)} - Y_{(L_2)}) \left[-L_1(R + H X_{(L_1)} H^\intercal) + A X_{(L_1)} H^\intercal\right] \\
&+ 2 Y_{(L_2)} \left[(L_2 - L_1)(R + H X_{(L_1)} H^\intercal) + A_{L_2}(X_{(L_1)} - X_{(L_2)}) H^\intercal\right].
\end{aligned}$$

Therefore,

$$\|\nabla J(L_1) - \nabla J(L_2)\|_F^2 \leq \ell_1^2 \|Y_{(L_1)} - Y_{(L_2)}\|_F^2 + \ell_2^2 \|L_1 - L_2\|_F^2 + \ell_3^2 \|X_{(L_1)} - X_{(L_2)}\|_F^2 \quad (25)$$

where

$$\begin{aligned}
\ell_1 = \ell_1(L_1) :=& 2\| - L_1(R + H X_{(L_1)} H^\intercal) + A X_{(L_1)} H^\intercal\|_{\mathrm{op}}, \\
\ell_2 = \ell_2(L_1, L_2) :=& 2\|Y_{(L_2)}\|_{\mathrm{op}} \|R + H X_{(L_1)} H^\intercal\|_{\mathrm{op}}, \\
\ell_3 = \ell_3(L_2) :=& 2\|Y_{(L_2)}\|_{\mathrm{op}} \|A_{L_2}\|_{\mathrm{op}} \|H^\intercal\|_{\mathrm{op}}.
\end{aligned}$$

On the other hand, by direct computation we obtain

$$\begin{aligned}
Y_{(L_1)} - Y_{(L_2)} &- A_{L_1}^\intercal (Y_{(L_1)} - Y_{(L_2)}) A_{L_1} \\
=& (L_2 - L_1)^\intercal H^\intercal Y_{(L_2)} A_{L_2} + A_{L_2}^\intercal Y_{(L_2)} H(L_2 - L_1) \\
&+ (L_1 - L_2)^\intercal H^\intercal Y_{(L_2)} H(L_1 - L_2) \\
\preceq& \|L_1 - L_2\|_F \, \ell_4 \, I
\end{aligned} \quad (26)$$

where

$$\ell_4 = \ell_4(L_1, L_2) := 2\|H^\intercal Y_{(L_2)} A_{L_2}\|_{\mathrm{op}} + \|H^\intercal Y_{(L_2)} H(L_1 - L_2)\|_{\mathrm{op}}.$$

Now, consider the mapping $L \to Z_{(L)}$ where $Z_{(L)} = Z$ is the unique solution of the following Lyapunov equation:

$$Z = A_L^\intercal Z A_L + I,$$

which is well-defined and continuous on $\mathcal{S} \supset \mathcal{S}_\alpha$. Therefore, by comparison, we claim that

$$\|Y_{(L_1)} - Y_{(L_2)}\|_F \preceq \|L_1 - L_2\|_F \, \ell_4 \, \|Z_{(L_1)}\|_F.$$

By a similar computation to that of (20), we obtain that

$$
\begin{aligned}
X_{(L_1)} - X_{(L_2)} &- A_{L_2}(X_{(L_1)} - X_{(L_2)})A_{L_2}^{\mathsf{T}} \\
&= (L_1 R - A_{L_1} X_{(L_1)} H^{\mathsf{T}})(L_1 - L_2)^{\mathsf{T}} \\
&\quad + (L_1 - L_2)(R L_1^{\mathsf{T}} - H X_{(L_1)} A_{L_1}^{\mathsf{T}}) \\
&\quad - (L_1 - L_2) R (L_1 - L_2)^{\mathsf{T}} \\
&\quad - (L_1 - L_2) H X_{(L_1)} H^{\mathsf{T}} (L_1 - L_2)^{\mathsf{T}} \\
&\preceq \|L_1 - L_2\|_F\, \ell_5\, (Q + L_2 R L_2^{\mathsf{T}})
\end{aligned}
\tag{27}
$$

where

$$
\ell_5 = \ell_5(L_1) := 2\| - L_1 R + A_{L_1} X_{(L_1)} H^{\mathsf{T}}\|_{\mathrm{op}}/\underline{\lambda}(Q).
$$

Therefore, by comparison, we claim that

$$
\|X_{(L_1)} - X_{(L_2)}\|_F \preceq \|L_1 - L_2\|_F\, \ell_5\, \|X_{(L_2)}\|_F.
$$

Finally, by compactness of $\mathcal{S}_\alpha$, we claim that the following supremums are attained and thus, are achieved with some *finite* positive values:

$$
\begin{aligned}
\bar{\ell}_1(\alpha) &:= \sup_{L_1, L_2 \in \mathcal{S}_\alpha} \ell_1(L_1)\ell_4(L_1, L_2)\, \|Z_{(L_1)}\|_F, \\
\bar{\ell}_2(\alpha) &:= \sup_{L_1, L_2 \in \mathcal{S}_\alpha} \ell_2(L_1, L_2), \\
\bar{\ell}_3(\alpha) &:= \sup_{L_1, L_2 \in \mathcal{S}_\alpha} \ell_3(L_2)\ell_5(L_1)\|X_{(L_2)}\|_F.
\end{aligned}
$$

Then, the claim follows by combining the bound in (25) with (26) and (27), and the following choice of

$$
\ell(\alpha) := \sqrt{\bar{\ell}_1^2(\alpha) + \bar{\ell}_2^2(\alpha) + \bar{\ell}_3^2(\alpha)}.
$$

## D.7  Gradient Flow (GF)

For completeness, in the next two sections, we anlayze first-order methods in order to solve the minimization problem (10), although they are not part of the main paper. In this section, we consider a policy update according to the the GF dynamics:

[GF] $\qquad\qquad \dot{L}_s = -\nabla J(L_s).$

We summarize the convergence result in the following proposition which is a direct consequence of Lemma 2.

**Proposition 6.** *Consider any sublevel set $\mathcal{S}_\alpha$ for some $\alpha > 0$. Then, for any initial policy $L_0 \in \mathcal{S}_\alpha$, the GF updates converges to optimality at a linear rate of $\exp(-c_1(\alpha))$ (in both the function value and the policy iterate). In particular, we have*

$$
J(L_s) - J(L^*) \le (\alpha - J(L^*)) \exp(-c_1(\alpha)s),
$$

*and*

$$
\|L_s - L^*\|_F^2 \le \frac{\alpha - J(L^*)}{c_3(\alpha)} \exp(-c_1(\alpha)s).
$$

*Proof.* Consider a Lyapunov candidate function $V(L) := J(L) - J(L^*)$. Under the GF dynamics

$$
\dot{V}(L_s) = -\langle \nabla J(L_s), \nabla J(L_s) \rangle \le 0.
$$

Therefore, $L_s \in \mathcal{S}_\alpha$ for all $s > 0$. But then, by Lemma 2, we can also show that

$$
\dot{V}(L_s) \le -c_1(\alpha) V(L_s) - c_2(\alpha)\|L_s - L^*\|_F^2, \quad \text{for } s > 0.
$$

By recalling that $c_1(\alpha) > 0$ is a positive constant independent of $L$, we conclude the following exponential stability of the GF:

$$
V(L_s) \le V(L_0) \exp(-c_1(\alpha)s),
$$

for any $L_0 \in \mathcal{S}_\alpha$ which, in turn, guarantees convergence of $J(L_s) \to J(L^*)$ at the linear rate of $\exp(-c_1(\alpha))$. Finally, the linear convergence of the policy iterates follows directly from the second bound in Lemma 2:

$$
\|L_s - L^*\|_F^2 \le \tfrac{1}{c_3(\alpha)} V(L_s) \le \tfrac{V(L_0)}{c_3(\alpha)} \exp(-c_1(\alpha)s).
$$

The proof concludes by noting that $V(L_0) \le \alpha - J(L^*)$ for any such initial value $L_0 \in \mathcal{S}_\alpha$. $\qquad\square$

## D.8 Gradient Descent (GD)

Here, we consider the GD policy update:

[GD] $\qquad L_{k+1} = L_k - \eta_k \nabla J(L_k),$

for $k \in \mathbb{Z}$ and a positive stepsize $\eta_k$. Given the convergence result for the GF, establishing convergence for GD relies on carefully choosing the stepsize $\eta_k$, and bounding the rate of change of $\nabla J(L)$—at least on each sublevel set. This is achieved by the Lipschitz bound for $\nabla J(L)$ on any sublevel set.

In what follows, we establish linear convergence of the GD update. Our convergence result only depends on the value of $\alpha$ for the initial sublevel set $\mathcal{S}_\alpha$ that contains $L_0$. Note that our proof technique is distinct from those in [23] and [50]; nonetheless, it involves a similar argument using the gradient dominance property of $J$.

**Theorem 4.** *Consider any sublevel set $\mathcal{S}_\alpha$ for some $\alpha > 0$. Then, for any initial policy $L_0 \in \mathcal{S}_\alpha$, the GD updates with any fixed stepsize $\eta_k = \eta \in (0, 1/\ell(\alpha)]$ converges to optimality at a linear rate of $1 - \eta c_1(\alpha)/2$ (in both the function value and the policy iterate). In particular, we have*

$$J(L_k) - J(L^*) \le [\alpha - J(L^*)](1 - \eta c_1(\alpha)/2)^k,$$

*and*

$$\|L_k - L^*\|_F^2 \le \left[ \frac{\alpha - J(L^*)}{c_3(\alpha)} \right] (1 - \eta c_1(\alpha)/2)^k,$$

*with $c_1(\alpha)$ and $c_3(\alpha)$ as defined in Lemma 2.*

*Proof.* First, we argue that the GD update with such a step size does not leave the initial sublevel set $\mathcal{S}_\alpha$ for any initial $L_0 \in \mathcal{S}_\alpha$. In this direction, consider $L(\eta) = L_0 - \eta \nabla J(L_0)$ for $\eta \ge 0$ where $L_0 \ne L^*$. Then, by compactness of $\mathcal{S}_\alpha$ and continuity of the mapping $\eta \to J(L(\eta))$ on $\mathcal{S} \supset \mathcal{S}_\alpha$, the following supremum is attained with a positive value $\bar\eta_0$:

$$\bar\eta_0 := \sup\{\eta : J(L(\zeta)) \le \alpha, \forall \zeta \in [0, \eta]\},$$

where positivity of $\bar\eta_0$ is a direct consequence of the strict decay of $J(L(\eta))$ for sufficiently small $\eta$ as $\nabla J(L_0) \ne 0$. This implies that $L(\eta) \in \mathcal{S}_\alpha \subset \mathcal{S}$ for all $\eta \in [0, \bar\eta_0]$ and $J(L(\bar\eta_0)) = \alpha$. Next, by the Fundamental Theorem of Calculus and smoothness of $J(\cdot)$ (Lemma 7), for any $\eta \in [0, \bar\eta_0]$ we have that,

$$J(L(\eta)) - J(L_0) - \langle \nabla J(L_0), L(\eta) - L_0 \rangle = \int_0^1 \langle \nabla J(L(\eta s)) - \nabla J(L_0), L(\eta) - L_0 \rangle ds$$

$$\le \|L(\eta) - L_0\|_F \int_0^1 \|\nabla J(L(\eta s)) - \nabla J(L_0)\|_F ds$$

$$\le \ell(\alpha)\|L(\eta) - L_0\|_F \int_0^1 \|L(\eta s) - L_0\|_F ds$$

$$= \frac{1}{2}\ell(\alpha)\eta\|L(\eta) - L_0\|_F \|\nabla J(L_0)\|_F,$$

where $\|\cdot\|_F$ denotes the Frobenius norm, the first inequality is a consequence of Cauchy-Schwartz, and the second one is due to (11c) and the fact that $L(\eta s)$ remains in $\mathcal{S}_\alpha$ for all $s \in [0, 1]$.[3] By the definition of $L(\eta)$, it now follows that,

$$J(L(\eta)) - J(L_0) \le \eta\|\nabla J(L_0)\|_F^2 \left( \frac{\ell(\alpha)\eta}{2} - 1 \right). \tag{28}$$

This implies $J(L(\eta)) \le J(L_0)$ for all $\eta \le 2/\ell(\alpha)$, and thus concluding that $\bar\eta_0 \ge 2/\ell(\alpha)$. This justifies that $L(\eta) \in \mathcal{S}_\alpha$ for all $\eta \in [0, 2/\ell(\alpha)]$. Next, if we consider the GD update with any

---

[3]Note that a direct application of Descent Lemma [51, Lemma 5.7] may not be justified as one has to argue about the uniform bound for the Hessian of $J$ over the non-convex set $\mathcal{S}_\alpha$ where $J$ is $\ell(\alpha)$-Lipschitz only on $\mathcal{S}_\alpha$. Also see the proof of [50, Theorem 2].

fixed stepsize $\eta \in (0, 1/\ell(\alpha)]$ and apply the bound in (28) and the gradient dominance property in Lemma 2, we obtain

$$J(L_1) - J(L_0) \le \eta c_1(\tfrac{\ell(\alpha)\eta}{2} - 1)[J(L_0) - J(L^*)],$$

which by subtracting $J(L^*)$ results in

$$J(L_1) - J(L^*) \le \left(1 - \tfrac{\eta c_1}{2}\right)[J(L_0) - J(L^*)],$$

as $\eta c_1(\ell(\alpha)\eta/2 - 1) \le -\eta c_1/2$ for all $\eta \in (0, 1/\ell(\alpha)]$. By induction, and the fact that both $c_1(\alpha)$ and the choice of $\eta$ only depends on the value of $\alpha$, we conclude the convergence in the function value at a linear rate of $1 - (\eta c_1/2)$ and the constant coefficient of $\alpha - J(L^*) \ge J(L_0) - J(L^*)$. To complete the proof, the linear convergence of the policy iterates follows directly from the second bound in Lemma 2. $\qquad\square$

## Appendix E  Proofs for the analysis of the constrained SGD algorithm

### E.1  Proof of lemma 3: Derivation of stochastic gradient formula

*Proof.* For small enough $\Delta \in \mathbb{R}^{n \times m}$,

$$\varepsilon(L + \Delta, \mathcal{Y}) - \varepsilon(L, \mathcal{Y}) = \|e_T(L + \Delta)\|^2 - \|e_T(L)\|^2$$
$$= 2\text{tr}\left[(e_T(L + \Delta) - e_T(L))e_T^\intercal(L)\right] + o(\|\Delta\|)).$$

The difference

$$e_T(L + \Delta) - e_T(L) = E_1(\Delta) + E_2(\Delta) + o(\|\Delta\|),$$

with the following terms that are linear in $\Delta$:

$$E_1(\Delta) := -\sum_{t=0}^{T-1} H(A_L)^t \Delta y(T - t - 1),$$
$$E_2(\Delta) := \sum_{t=1}^{T-1}\sum_{k=1}^{t} H(A_L)^{t-k} \Delta H(A_L)^{k-1} Ly(T - t - 1).$$

Therefore, combining the two identities, the definition of gradient under the inner product $\langle A, B \rangle := \text{tr}[AB^\intercal]$, and ignoring the higher order terms in $\Delta$ yields,

$$\langle \nabla_L \varepsilon(L, y), \Delta \rangle = 2\text{tr}\left[(E_1(\Delta) + E_2(\Delta))e_T^\intercal(L)\right],$$

which by linearity and cyclic permutation property of trace reduces to:

$$\langle \nabla_L \varepsilon(L, y), \Delta \rangle = -2\text{tr}\left[\Delta\left(\sum_{t=0}^{T-1} y(T - t - 1)e_T^\intercal(L)H(A_L)^t\right)\right]$$
$$+ 2\text{tr}\left[\Delta\left(\sum_{t=1}^{T-1}\sum_{k=1}^{t} H(A_L)^{k-1} Ly(T - t - 1)e_T^\intercal(L)H(A_L)^{t-k}\right)\right].$$

This holds for all admissible $\Delta$, concluding the formula for the gradient. $\qquad\square$

### E.2  Proof of Lemma 4: Robustness of the policy with respect to perturbation

*Proof.* Recall the stability certificate $s_K$ as proposed in [30, Lemma IV.1] for a choice of constant mapping $\mathcal{Q} : K \to \Lambda \succ 0$ and dual problem parameters as discussed in Remark 3. Then, we arrive at $s_K = \underline{\lambda}(\Lambda)/(2\,\overline{\lambda}(Z)\|H^\intercal\Delta^\intercal\|)$, for which $\rho\left(A^\intercal + H^\intercal(L^\intercal + \eta\Delta^\intercal)\right) < 1$ for any $\eta \in [0, s_K]$. But, the spectrum of a square matrix and its transpose are identical, thus $L + \eta\Delta \in \mathcal{S}$ for any such $\eta$. The claim then follows by noting that the operator norm of a matrix and its transpose are identical, and the resulting lowerbound as follows $s_K \ge \underline{\lambda}(\Lambda)/(2\,\overline{\lambda}(Z)\|H\|\|\Delta\|_F)$. $\qquad\square$

### E.3  Proof of Lemma 5: Uniform lower-bound on stepsize

*Proof.* Without loss of generality, suppose $L_0 \ne L^*$ and $E \ne 0$, and let $L(\eta) := L_0 - \eta E$. By compactness of $\mathcal{S}_\beta$ and continuity of the mapping $\eta \to J(L(\eta))$ on $\mathcal{S} \supset \mathcal{S}_\beta$, the following supremum is attained by $\eta_\beta$:

$$\eta_\beta := \sup\{\eta : J(L(\zeta)) \le \beta, \forall \zeta \in [0, \eta]\}. \tag{29}$$

Note that $\eta_\beta$ is strictly positive for $\beta > \alpha$ because $L_0 \in \mathcal{S}_\alpha \subset \mathcal{S}_\beta$ and $J(\cdot)$ is coercive (Lemma 1) and its domain is open (Lemma 4). This implies that $L(\eta) \in \mathcal{S}_\beta \subset \mathcal{S}$ for all $\eta \in [0, \eta_\beta]$ and $J(L(\eta_\beta)) = \beta$.

Next, we want to show that $\eta_\beta$ is uniformly lower bounded with high probability. By the Fundamental Theorem of Calculus, for any $\eta \in [0, \eta_\beta]$ we have

$$
\begin{aligned}
J(L(\eta)) - J(L_0) - \langle \nabla J(L_0), L(\eta) - L_0 \rangle &= \int_0^1 \langle \nabla J(L(\eta s)) - \nabla J(L_0), L(\eta) - L_0 \rangle ds \\
&\leq \|L(\eta) - L_0\|_F \int_0^1 \|\nabla J(L(\eta s)) - \nabla J(L_0)\|_F ds \\
&\leq \ell(\beta) \|L(\eta) - L_0\|_F \int_0^1 \|L(\eta s) - L_0\|_F ds \\
&= \frac{1}{2} \ell(\beta) \eta^2 \|E\|_F^2, \quad (30)
\end{aligned}
$$

where the first inequality is a consequence of Cauchy-Schwartz, and the second one is due to (11c) and the fact that $L(\eta s)$ remains in $\mathcal{S}_\beta$ for all $s \in [0, 1]$ by definition of $\eta_\beta$. Note that the assumption implies $\|E\|_F \leq (\gamma + 1)\|\nabla J(L_0)\|_F$. Thus, (30) implies that

$$
\begin{aligned}
J(L(\eta)) \leq &J(L_0) - \eta \langle \nabla J(L_0), E \rangle + \frac{1}{2} \ell(\beta) \eta^2 \|E\|_F^2 \\
\leq &J(L_0) - \eta \langle \nabla J(L_0), E - \nabla J(L_0) \rangle - \eta \|\nabla J(L_0)\|_F^2 + \frac{1}{2} \ell(\beta) \eta^2 \|E\|_F^2 \quad (31) \\
\leq &J(L_0) + \|\nabla J(L_0)\|_F^2 \left[ \frac{1}{2} (\gamma + 1)^2 \ell(\beta) \eta^2 + (\gamma - 1)\eta \right].
\end{aligned}
$$

Therefore, for $\eta$ to be a feasible point in the supremum in (29), it suffices to satisfy:

$$
\frac{1}{2} \|\nabla J(L_0)\|_F^2 \left[ (\gamma + 1)^2 \ell(\beta) \eta^2 + 2(\gamma - 1)\eta \right] \leq \beta - \alpha,
$$

or equivalently,

$$
\left[ (\gamma + 1)\ell(\beta)\eta + \frac{\gamma - 1}{\gamma + 1} \right]^2 \leq \left( \frac{\gamma - 1}{\gamma + 1} \right)^2 + \frac{2\ell(\beta)[\beta - \alpha]}{\|\nabla J(L_0)\|_F^2}.
$$

But then it suffices to have

$$
(\gamma + 1)\ell(\beta)\eta + \frac{\gamma - 1}{\gamma + 1} \leq \frac{\sqrt{2\ell(\beta)[\beta - \alpha]}}{\|\nabla J(L_0)\|_F}.
$$

Finally, note that by Lemma 2 and (11c) we have

$$
\|\nabla J(L_0)\|_F \leq \frac{\ell(\alpha)}{c_3(\alpha)} [J(L_0) - J(L^*)] \leq \frac{\ell(\alpha)}{c_3(\alpha)} [\alpha - \alpha^*]. \quad (32)
$$

Using this uniform bound of gradient on sublevel set $\mathcal{S}_\alpha$ and noting that $\gamma \in [0, 1]$, we can obtain the sufficient condition for $\eta$ to be feasible. This completes the proof. □

### E.4 Proof of Proposition 2: Linear decay in cost value

*Proof.* Suppose $L_0 \neq L^*$ and let $L(\eta) := L_0 - \eta E$. Note that $E$ may not be necessarily in the direction of decay in $J(L)$, however, we can argue the following:

Choose $\beta = \alpha$ in Lemma 5 and note that $\eta_\beta$ as defined in (29) will be lower bounded as $\eta_\beta \geq \bar{\eta}_0$, and thus $\bar{\eta}_0$ is feasible. Recall that $L(\eta) \in \mathcal{S}_\beta \subset \mathcal{S}$ for all $\eta \in [0, \eta_\beta]$. Also, for any $\eta \in [0, \bar{\eta}_0]$ and $\gamma \in [0, 1)$, from (31) we obtain that:

$$
\begin{aligned}
J(L(\eta)) - J(L_0) &\leq \|\nabla J(L_0)\|_F^2 \left[ \frac{1}{2}(\gamma + 1)^2 \ell(\alpha)\eta^2 + (\gamma - 1)\eta \right] \\
&\leq c_1(\alpha)[J(L_0) - J(L^*)] \left[ \frac{1}{2}(\gamma + 1)^2 \ell(\alpha)\eta^2 + (\gamma - 1)\eta \right]
\end{aligned}
$$

where, as $\gamma < 1$, the last inequality follows by (11a) for any $\eta \leq \min\{2\bar{\eta}_0, \eta_\beta\}$. By the choice of $\bar{\eta}_0$, then we obtain that

$$J(L(\bar{\eta}_0)) - J(L_0) \leq -c_1(\alpha) \left[ \frac{(\gamma - 1)^2}{2(\gamma + 1)^2 \ell(\alpha)} \right] [J(L_0) - J(L^*)]$$

This implies that

$$J(L(\bar{\eta}_0)) - J(L^*) \leq \left( 1 - c_1(\alpha) \left[ \frac{(\gamma - 1)^2}{2(\gamma + 1)^2 \ell(\alpha)} \right] \right) [J(L_0) - J(L^*)].$$

$\square$

### E.5 Proof of Theorem 1: Convergence of SGD algorithm

Before proving this result, we first discuss that the claim of Theorem 1 is sufficient for establishing the complexity result of Theorem 3; i.e., it suffices to guarantee convergence from every initial (stabilizing) policy to a neighborhood of optimality where norm of the gradient is controlled.

As shown in Lemma 2 (and discussed in Remark 4), the cost maintains the PL property on each sublevel set. In particular (11a) implies that on each $\mathcal{S}_\alpha$, $\|\nabla J(L)\|$ characterizes the optimality gap $J(L) - J(L^*)$ by:

$$c_1(\alpha)[J(L) - J(L^*)] \leq \|\nabla J(L)\|^2$$

for some constant $c_1(\alpha)$. This implies that if we have arrived at a candidate policy $L_k$ for which the gradient is small, then the optimality gap should be small (involving the constant $c_1(\alpha)$ that is independent of $L_k$). This is the reason that in Theorem 1, it suffices to argue about the generated sequence $L_k$ to have a linear decay unless entering a neighborhood of $L^*$ containing policies with small enough gradients (denoted by $\mathcal{C}_\tau$). In particular, if for some $j < k$, we arrive at some policy $L_j \in \mathcal{C}_\tau$, then by (11a) we can conclude that:

$$J(L_j) - J(L^*) \leq \frac{1}{c_1(\alpha)} \|\nabla J(L_j)\|_F^2 \leq \frac{s_0^2}{c_1(\alpha)\tau^2},$$

which is directly controlled by the bias term $s_0$. This is the bound used also in the proof of Theorem 3.

Finally, recall that every (stabilizing) initial policy $L_0$ amounts to a *finite* value of $J(L_0)$ and thus lies in some sublevel set $S_\alpha$. So, starting from such $L_0$, Theorem 1 guarantees linear decay of the optimality gap till the trajectory enters that small neighborhood. Finally, note that the radius of this neighborhood $\mathcal{C}_\tau$ is characterized by the bias term $s_0$ which itself is exponentially decaying to zero in the trajectory length $T$.

Next, we provide the proof of this result that is essentially an induction argument using Proposition 2.

*Proof of Theorem 1.* The first step of the proof is to show that the assumption of Proposition 2 is satisfied for $E = \nabla\widehat{J}(L)$ for all $L \in \mathcal{S}_\alpha \setminus \mathcal{C}_{\gamma/2}$. This is true because

$$\|\nabla\widehat{J}(L) - \nabla J(L)\| \leq s\|\nabla J(L)\| + s_0$$
$$\leq s\|\nabla J(L)\| + \frac{\gamma}{2}\|\nabla J(L)\|$$
$$\leq \gamma\|\nabla J(L)\|$$

where the first inequality follows from Assumption 3, the second inequality follows from $L \notin C_{\gamma/2}$ (i.e. $s_0\|\nabla J(L)\| \geq \gamma/2$), and the last step follows from the assumption $s \leq \gamma/2$.
The rest of the proof relies on repeated application of Proposition 2. In particular, starting from $L_0 \in \mathcal{S}_\alpha \setminus \mathcal{C}_{\gamma/2}$, the application of Proposition 2 implies that $L_1 := L_0 - \bar{\eta}\nabla\widehat{J}(L_0)$ remains in the same sublevel set, i.e., $L_1 \in \mathcal{S}_\alpha$, and we obtain the following linear decay of the cost value:

$$J(L_1) - J(L^*) \leq [1 - c_1(\alpha)\bar{\eta}(1 - \gamma)/2] [J(L_0) - J(L^*)].$$

Now, if $L_1 \in \mathcal{C}_{\gamma/2}$ then we stop; otherwise $L_1 \in \mathcal{S}_\alpha \setminus \mathcal{C}_{\gamma/2}$ and we can repeat the above process to arrive at $L_2 := L_1 - \bar{\eta}\nabla\widehat{J}(L_1) \in \mathcal{S}_\alpha$, with a guaranteed linear decay

$$J(L_2) - J(L^*) \leq [1 - c_1(\alpha)\bar{\eta}(1 - \gamma)/2] [J(L_1) - J(L^*)].$$

Combining the last two linear decays yields

$$J(L_2) - J(L^*) \leq [1 - c_1(\alpha)\bar{\eta}(1 - \gamma)/2]^2 [J(L_0) - J(L^*)].$$

Repeating the process generates a sequence of policies $L_0, L_1, L_2...$ with a combined linear decay of

$$J(L_k) - J(L^*) \leq [1 - c_1(\alpha)\bar{\eta}(1 - \gamma)/2]^k [J(L_0) - J(L^*)],$$

unless at some iteration $j$, we arrive at a policy $L_j$ such that $L_j \in \mathcal{C}_{\gamma/2}$. This completes the proof. $\square$

## Appendix F  Proofs of the result for observation model and sample complexity

### F.1  Preliminary lemmas and their proofs

First, we provide the proof for the complete version of Lemma 6:[4]

**Lemma 6'** (Uniform Bounds for Stable Systems). *Suppose $L \in \mathcal{S}$, then there exit a constants $C_L > 0$ such that*

$$\|A_L^k\| \leq C_L \left(\sqrt{\rho(A_L)}\right)^{k+1}, \quad \forall k \geq 0,$$

*whenever $\rho(A_L) > 0$, and otherwise $\sqrt{\rho(A_L)}$ is replaced with any arbitrarily small $r \in (0,1)$. Additionally,*

$$\sum_{i=0}^{\infty} \|A_L^i\| \leq \frac{C_L}{1 - \sqrt{\rho(A_L)}}$$

$$\sum_{i=0}^{\infty} \|M_i[E]\| \leq \frac{1 + 2C_L^2\rho(A_L)^{3/2}}{[1 - \sqrt{\rho(A_L)}]^2} \|EH\|$$

*Furthermore, consider $\mathcal{S}_\alpha$ for some $\alpha > 0$, then there exist constants $D_\alpha > 0$, $C_\alpha > 0$ and $\rho_\alpha \in (0,1)$ such that $\|L\| \leq D_\alpha$, $C_L \leq C_\alpha$ and $\rho(A_L) \leq \rho_\alpha$, $\forall L \in \mathcal{S}_\alpha$.*

*Proof of Lemma 6.* Recall the Cauchy Integral formula for matrix functions [53, Theorem 1.12]: for any matrix $M \in \mathbb{C}^{n \times n}$,

$$f(M) = \frac{1}{2\pi i} \oint_\Gamma f(z)(zI - M)^{-1}dz,$$

whenever $f$ is real analytic on and inside a closed contour $\Gamma$ that encloses spectrum of $M$. Note that $L \in \mathcal{S}$ implying that $\rho(A_L) < 1$. Now, fix some $r \in (\rho(A_L), 1)$ and define $\Gamma(\theta) = re^{i\theta}$ with $\theta$ ranging on $[0, 2\pi]$. Therefore, Cauchy Integral formula applies to $f(z) = z^k$ for any positive integer $k$ and the contour $\Gamma$ defined above. So, for matrix $A_L$, we obtain

$$A_L^k = \frac{1}{2\pi i} \oint_\Gamma z^k(zI - A_L)^{-1}dz = \frac{1}{2\pi i} \int_0^{2\pi} r^k e^{ik\theta}(re^{i\theta}I - A)^{-1} r\, de^{i\theta},$$

implying that

$$\|A_L^k\| \leq \frac{r^{k+1}}{2\pi} \int_0^{2\pi} \|(re^{i\theta}I - A_L)^{-1}\|d\theta \leq r^{k+1} \max_{\theta \in [0,2\pi]} \|(re^{i\theta}I - A_L)^{-1}\|.$$

Finally, the first claim follows by choosing $r = \sqrt{\rho(A_L)}$ (whenever $\rho(A_L) > 0$, otherwise $r \in (0,1)$ can be chosen arbitrarily small) and defining

$$C_L := \max_{\theta \in [0,2\pi]} \|(\sqrt{\rho(A_L)}e^{i\theta}I - A_L)^{-1}\|$$

which is attained and bounded.

---

[4]See [52, Definition 3.1] for an alternative notation; however, we prefer the explicit and simple form of expressing spectral radius in the bounds established in our work, which also facilitates the comparison to literature on first order methods for stabilizing policies.

Next, by applying the first claim, we have

$$\sum_{i=0}^{T} \|A_L^i\| \le C_L \frac{1 - \sqrt{\rho(A_L)}^T}{1 - \sqrt{\rho(A_L)}},$$

implying the second bound. For the third claim, note that for $i = 1, 2, \cdots$ we obtain

$$\|M_{i+1}[E]\| \le \sum_{k=0}^{i} \|A_L^{i-k}\| \|A_L^k\| \|EH\| \le \|EH\| C_L^2 \sum_{k=0}^{i} \left[\sqrt{\rho(A_L)}\right]^{(i-k+1)+(k+1)}$$

$$\le \|EH\| C_L^2 \sqrt{\rho(A_L)} \left[(i+1) \cdot \rho(A_L)^{(i+1)/2}\right]. \quad (33)$$

But, then by recalling that $M_0[E] = 0$ and $\|M_1[E]\| = \|EH\|$ we have

$$\sum_{i=0}^{\infty} \|M_i[E]\| \le \|EH\| + \|EH\| \left[\frac{2C_L^2 \rho(A_L)^{3/2}}{[1 - \sqrt{\rho(A_L)}]^2}\right]$$

where we used the following convergent sum for any $\rho \in (0, 1)$:

$$\sum_{i=1}^{\infty} (i+1) \cdot \rho^{i+1} = \frac{(2-\rho)\rho^2}{(1-\rho)^2} \le \frac{2\rho^2}{(1-\rho)^2}.$$

This implies the third bound.

The final claim follows directly from compactness of sublevel set $\mathcal{S}_\alpha$ (Lemma 1) and continuity of the mappings $(L, \theta) \mapsto (\rho(A_L), \theta) \mapsto \|(\sqrt{\rho(A_L)}e^{i\theta}I - A_L)^{-1}\|$ on $\mathcal{S}_\alpha \times [0, 2\pi]$ whenever $\rho(A_L) > 0$ (and otherwise considering the mapping $(L, \theta) \mapsto \|(re^{i\theta}I - A_L)^{-1}\|$ for arbitrarily small and fixed $r \in (0, 1)$). $\qquad\square$

As mentioned in Section 4.2, a key idea behind these error bounds that scale well with respect to the length T is the following consequence of von Neumann Trace Inequality [49, Theorem 8.7.6]:

$$|\mathrm{tr}\,[MN]| \le \textstyle\sum_{i=1}^{T} \sigma_i(M)\sigma_i(N) \le \|M\|\|N\|_*,$$

with $\|\cdot\|_*$ denoting the nuclear norm. Additionally, as a direct consequence of Courant-Fischer Theorem, one can also show that nuclear norm is sub-multiplicative. More precisely,

$$\|AB\|_* \le \|A\| \|B\|_* \le \|A\|_* \|B\|_*.$$

Next, we require the following lemma to bound these errors.

**Lemma 8.** *For any $L \in \mathcal{S}_\alpha$, we have*

$$\|\mathcal{A}_L^\intercal H^\intercal H \mathcal{A}_L\|_* \le \frac{C_L^2 \|H^\intercal H\|_*}{1 - \rho(A_L)},$$

$$\|\mathcal{N}_L[E]\|_* \le \frac{\left[2C_L + 4C_L^3 \rho(A_L)^{3/2}\right] \|H\| \|H^\intercal H\|_*}{[1 - \rho(A_L)]^2} \|E\|.$$

*Proof of Lemma 8.* For the first claim, note that $\mathcal{A}_L^\intercal H^\intercal H \mathcal{A}_L$ is positive semi-definite, so

$$\|\mathcal{A}_L^\intercal H^\intercal H \mathcal{A}_L\|_* = \mathrm{tr}\,[\mathcal{A}_L^\intercal H^\intercal H \mathcal{A}_L]$$

$$\le \mathrm{tr}\,[H^\intercal H] \|\mathcal{A}_L \mathcal{A}_L^\intercal\|$$

$$\le \|H^\intercal H\|_* \left\|\sum_{i=0}^{T} A_L^i (A_L^\intercal)^i\right\|$$

$$\le \|H^\intercal H\|_* \sum_{i=0}^{T} \|A_L^i\|^2$$

$$\le \|H^\intercal H\|_* \frac{C_L^2}{1 - \rho(A_L)}$$

where the last inequality follows by Lemma 6. Next, we have

$$\|\mathcal{M}_L[E]\| = \|\mathcal{M}_L[E]\mathcal{M}_L[E]^{\mathsf{T}}\|^{1/2}$$

$$\leq \left[\sum_{i=0}^{T} \|M_i[E]\|^2\right]^{1/2}$$

$$\leq \|EH\| + \|EH\| C_L^2 \sqrt{\rho(A_L)} \left[\sum_{i=0}^{T}(i+1)^2 \cdot \rho(A_L)^{(i+1)}\right]^{1/2}$$

$$\leq \|EH\| + \|EH\| C_L^2 \sqrt{\rho(A_L)} \frac{2\rho(A_L)}{[1-\rho(A_L)]^{3/2}}$$

$$\leq \|EH\| \left[\frac{1 + 2C_L^2\rho(A_L)^{3/2}}{[1-\rho(A_L)]^{3/2}}\right]$$

where the second inequality follows by (33) and the third one by the following convergent sum for any $\rho \in (0,1)$:

$$\sum_{i=1}^{\infty}(i+1)^2 \cdot \rho^{i+1} = \frac{\rho^2(\rho^2 - 3\rho + 4)}{(1-\rho)^3} \leq \frac{4\rho^2}{(1-\rho)^3}.$$

Also, by the properties of nuclear norm

$$\|H^{\mathsf{T}}H\mathcal{A}_L\|_* = \mathrm{tr}\left[\sqrt{H^{\mathsf{T}}H\mathcal{A}_L\mathcal{A}_L^{\mathsf{T}}H^{\mathsf{T}}H}\right]$$

$$\leq \|\mathcal{A}_L\mathcal{A}_L^{\mathsf{T}}\|^{1/2}\|H^{\mathsf{T}}H\|_*$$

$$\leq \left[\sum_{i=0}^{\infty}\|A_L^i\|^2\right]^{1/2}\|H^{\mathsf{T}}H\|_*$$

$$\leq \left[\frac{C_L^2}{1-\rho(A_L)}\right]^{1/2}\|H^{\mathsf{T}}H\|_*,$$

where the last inequality follows by Lemma 6. Finally, notice that

$$\|\mathcal{N}_L[E]\|_* \leq 2\|\mathcal{A}_L^{\mathsf{T}}H^{\mathsf{T}}H\mathcal{M}_L[E]\|_*$$
$$\leq 2\|\mathcal{M}_L[E]\|\|H^{\mathsf{T}}H\mathcal{A}_L\|_*$$

and thus combining the last three bounds implies the second claim. This completes the proof. □

The next tool we will be using is the following famous bound on random matrices which is a variant of Bernstein inequality:

**Lemma 9** (Matrix Bernstein Inequality [54, Corollary 6.2.1]). *Let $Z$ be a $d_1 \times d_2$ random matrices such that $\mathbb{E}[Z] = \bar{Z}$ and $\|Z\| \leq K$ almost surely. Consider $M$ independent copy of $Z$ as $Z_1, \cdots, Z_M$, then for every $t \geq 0$, we have*

$$\mathbb{P}\left[\left\|\frac{1}{M}\sum_i Z_i - \bar{Z}\right\| \geq t\right] \leq (d_1 + d_2)\exp\left\{\frac{-Mt^2/2}{\sigma^2 + 2Kt/3}\right\}$$

*where $\sigma^2 = \max\{\|\mathbb{E}[ZZ^{\mathsf{T}}]\|, \|\mathbb{E}[Z^{\mathsf{T}}Z]\|\}$ is the per-sample second moment. This bound can be expressed as the mixture of sub-gaussian and sub-exponential tail as $(d_1 + d_2)\exp\left\{-c\min\{\frac{t^2}{\sigma^2}, \frac{t}{2K}\}\right\}$ for some $c$.*

We are now well-equipped to provide the main proofs.

### F.2 Proof of Proposition 3: The Observation model

*Proof.* Recall that

$$\varepsilon(L, \mathcal{Y}_T) = \|Hx(T) - H\hat{x}(T)\|^2 = \sum_{i=1}^{m} |H_i^\mathsf{T} x(T) - H_i^\mathsf{T} \hat{x}(T)|^2$$

where $H_i^\mathsf{T}$ is the $i$-th row of $H$. Also, by duality, if $z(t) = (A_L^\mathsf{T})^{T-t} H_i$ is the adjoint dynamics' closed-loop trajectory with control signal $u(t) = L^\mathsf{T} z(t)$ then

$$H_i^\mathsf{T} x(T) - H_i^\mathsf{T} \hat{x}(T) = \vec{z}_i^\mathsf{T} \vec{\xi} - \vec{u}^\mathsf{T} \vec{\omega}$$

where

$$\vec{z}_i^\mathsf{T} = \begin{pmatrix} z(T)^\mathsf{T} & z(T-1)^\mathsf{T} & \ldots & z(1)^\mathsf{T} & z(0)^\mathsf{T} \end{pmatrix},$$
$$\vec{u}^\mathsf{T} = \begin{pmatrix} u(T)^\mathsf{T} & u(T-1)^\mathsf{T} & \ldots & u(1)^\mathsf{T} & 0_m^\mathsf{T} \end{pmatrix}.$$

But $\vec{u} = (I \otimes L^\mathsf{T}) \vec{z}_i$ and then $\vec{z}_i = \mathcal{A}_L^\mathsf{T} H_i$. Therefore,

$$\varepsilon(L, \mathcal{Y}_T) = \sum_{i=1}^{m} |H_i^\mathsf{T} x(T) - H_i^\mathsf{T} \hat{x}(T)|^2$$
$$= \sum_{i=1}^{m} \operatorname{tr}\left[\vec{\xi}\vec{\xi}^\mathsf{T} \vec{z}_i \vec{z}_i^\mathsf{T}\right] - \operatorname{tr}\left[\left(\vec{\xi}\vec{\omega}^\mathsf{T}(I \otimes L^\mathsf{T}) + (I \otimes L)\vec{\omega}\vec{\xi}^\mathsf{T}\right)\vec{z}_i\vec{z}_i^\mathsf{T}\right]$$
$$+ \operatorname{tr}\left[\vec{\omega}\vec{\omega}^\mathsf{T}(I \otimes L^\mathsf{T})\vec{z}_i\vec{z}_i^\mathsf{T}(I \otimes L)\right]$$

Then, by using the fact that $\sum_{i=1}^{m} H_i H_i^\mathsf{T} = H^\mathsf{T} H$, we obtain that

$$\varepsilon(L, \mathcal{Y}_T) = \operatorname{tr}\left[\mathcal{X}_L \mathcal{A}_L^\mathsf{T} H^\mathsf{T} H \mathcal{A}_L\right].$$

Thus, we can rewrite the estimation error as

$$\varepsilon(L, \mathcal{Y}_T) = \left\langle \vec{\xi}, \vec{\xi} \right\rangle_{\mathcal{A}_L} + \left\langle (I \otimes L)\vec{\omega}, (I \otimes L)\vec{\omega} \right\rangle_{\mathcal{A}_L} - 2\left\langle \vec{\xi}, (I \otimes L)\vec{\omega} \right\rangle_{\mathcal{A}_L} = \|\vec{\eta}_L\|^2_{\mathcal{A}_L}$$

Next, we can compute that for small enough $E$

$$\mathcal{A}_{L+E} - \mathcal{A}_L = \mathcal{M}_L[E] + o(\|E\|).$$

This implies that

$$\mathrm{d}(\mathcal{A}_L^\mathsf{T} H^\mathsf{T} H \mathcal{A}_L)\big|_L [E] = \mathcal{M}_L[E]^\mathsf{T} H^\mathsf{T} H \mathcal{A}_L + \mathcal{A}_L^\mathsf{T} H^\mathsf{T} H \mathcal{M}_L[E]$$

On the other hand,

$$\mathcal{X}_{L+E} - \mathcal{X}_L = (I \otimes E)\vec{\omega}\vec{\omega}^\mathsf{T}(I \otimes L^\mathsf{T})$$
$$+ (I \otimes L)\vec{\omega}\vec{\omega}^\mathsf{T}(I \otimes E^\mathsf{T})$$
$$- \vec{\xi}\vec{\omega}^\mathsf{T}(I \otimes E^\mathsf{T}) + (I \otimes E)\vec{\omega}\vec{\xi}^\mathsf{T} + o(\|E\|).$$

Therefore, the second claim follows by the chain rule. $\qquad\square$

### F.3 Proof of Proposition 4: Concentration bounds

We provide the proof for a detailed version of Proposition 4:

**Proposition 4'** (Concentration independent of length $T$). *Consider length $T$ trajectories* $\{\mathcal{Y}^i_{[t_0, t_0+T]}\}_{i=1}^{M}$ *and let* $\widehat{J}_T(L) := \frac{1}{M} \sum_{i=1}^{M} \varepsilon(L, \mathcal{Y}^i_T)$. *Then, under Assumption 4, for any* $s > 0$

$$\mathbb{P}\left[|\widehat{J}_T(L) - J_T(L)| \leq s\right] \geq 1 - 2n \exp\left[\frac{-Ms^2/2}{\mu_L^2 + 2\mu_L s/3}\right],$$

$$\mathbb{P}\left[\|\nabla\widehat{J}_T(L) - \nabla J_T(L)\| \leq s\right] \geq 1 - 2n \exp\left[\frac{-Ms^2/2}{\nu_L^2 + 2\nu_L s/3}\right]$$

*where $\kappa_L = \kappa_\xi + \|L\|\kappa_\omega$ and*

$$\mu_L := \frac{\kappa_L^2 C_L^2}{[1 - \sqrt{\rho(A_L)}]^2} \|H^\mathsf{T} H\|_*$$

$$\nu_L := \frac{2\kappa_L \kappa_\omega C_L^2 + \left[C_L + 2C_L^3 \rho(A_L)^{3/2}\right] \|H\|\kappa_L^2}{[1 - \sqrt{\rho(A_L)}]^3} \|H^\mathsf{T} H\|_*.$$

*Proof of Proposition 4.* Note that $\mathbb{E}\left[\vec{\xi}\vec{\xi}^\mathsf{T}\right] = \begin{pmatrix} I \otimes Q & 0 \\ 0 & P_0 \end{pmatrix} =: \mathcal{Q}$, $\mathbb{E}\left[\vec{\omega}\vec{\omega}^\mathsf{T}\right] = \begin{pmatrix} I \otimes R & 0 \\ 0 & 0_m \end{pmatrix} =:$ $\mathcal{R}$, and $\mathbb{E}\left[\vec{\xi}\vec{\omega}^\mathsf{T}\right] = 0$. Assume $m_0 = 0$ and recall that $\langle \nabla\varepsilon(L, \mathcal{Y}_T), E\rangle = \mathrm{d}\,\varepsilon(\cdot, \mathcal{Y}_T)\big|_L (E)$ thus, using Proposition 3, we can rewrite the $J_T(L)$ and its gradient as

$$J_T(L) = \mathbb{E}\left[\varepsilon(L, \mathcal{Y}_T)\right] = \mathbb{E}\left[\|\vec{\eta}_L\|_{\mathcal{A}_L}^2\right] = \mathrm{tr}\left[\mathbb{E}\left[\mathcal{X}_L\right] \mathcal{A}_L^\mathsf{T} H^\mathsf{T} H \mathcal{A}_L\right]$$

where $\mathbb{E}\left[\mathcal{X}_L\right] = \mathcal{Q} + (I \otimes L)\mathcal{R}(I \otimes L^\mathsf{T})$. Therefore, by definition of $\widehat{J}_T(L)$ we obtain

$$\widehat{J}_T(L) = \mathrm{tr}\left[\mathcal{Z}_L \mathcal{A}_L^\mathsf{T} H^\mathsf{T} H \mathcal{A}_L\right]$$

with $\mathcal{Z}_L = \frac{1}{M}\sum_{i=1}^M \mathcal{X}_L(\mathcal{Y}^i)$ which can be expanded as

$$\mathcal{Z}_L = \mathcal{Z}_1 + (I \otimes L)\mathcal{Z}_2(I \otimes L^\mathsf{T}) - \mathcal{Z}_3(I \otimes L^\mathsf{T}) - (I \otimes L)\mathcal{Z}_3^\mathsf{T},$$

where

$$\mathcal{Z}_1 = \frac{1}{M}\sum_{i=1}^M \vec{\xi}_i \vec{\xi}_i^\mathsf{T}, \quad \mathcal{Z}_2 = \frac{1}{M}\sum_{i=1}^M \vec{\omega}_i \vec{\omega}_i^\mathsf{T}, \quad \mathcal{Z}_3 = \frac{1}{M}\sum_{i=1}^M \xi_i \vec{\omega}_i^\mathsf{T},$$

and $\mathbb{E}\left[\mathcal{Z}_L\right] = \mathcal{Q} + (I \otimes L)\mathcal{R}(I \otimes L^\mathsf{T})$. Therefore,

$$\widehat{J}_T(L) - J_T(L) = \mathrm{tr}\left[(\mathcal{Z}_L - \mathbb{E}\left[\mathcal{Z}_L\right]) \mathcal{A}_L^\mathsf{T} H^\mathsf{T} H \mathcal{A}_L\right].$$

Thus, by cyclic permutation property of trace and (12) we obtain

$$|\widehat{J}_T(L) - J_T(L)| \le \|\mathcal{A}_L (\mathcal{Z}_L - \mathbb{E}\left[\mathcal{Z}_L\right]) \mathcal{A}_L^\mathsf{T}\| \|H^\mathsf{T} H\|_*. \tag{34}$$

Next, we consider the symmetric random matrix $\mathcal{A}_L (\mathcal{Z}_L - \mathbb{E}\left[\mathcal{Z}_L\right]) \mathcal{A}_L^\mathsf{T}$. Note that $\|\xi(t) - L\omega(t)\| \le \kappa_L$ almost surely and thus

$$\|\mathcal{A}_L \mathcal{X}_L \mathcal{A}_L^\mathsf{T}\| = \|\mathcal{A}_L \vec{\eta}_L\|^2 \le \kappa_L^2 \left[\sum_{i=0}^\infty \|A_L^i\|\right]^2 \le \mu_L/\|H^\mathsf{T} H\|_*.$$

It then follows that

$$\left\|\mathbb{E}\left[(\mathcal{A}_L \mathcal{X}_L \mathcal{A}_L^\mathsf{T})^2\right]\right\| \le \mathbb{E}\left[\|\mathcal{A}_L \mathcal{X}_L \mathcal{A}_L^\mathsf{T}\|^2\right] \le \mu_L^2/\|H^\mathsf{T} H\|_*^2.$$

Therefore, by Lemma 9 we obtain that

$$\mathbb{P}\left[\|\mathcal{A}_L (\mathcal{Z}_L - \mathbb{E}\left[\mathcal{Z}_L\right]) \mathcal{A}_L^\mathsf{T}\| \ge t\right] \le 2n \exp\left[\frac{-M\|H^\mathsf{T} H\|_*^2 t^2/2}{\mu_L^2 + 2\mu_L\|H^\mathsf{T} H\|_* t/3}\right].$$

Substituting $t$ with $t/\|H^\mathsf{T} H\|_*$ together with (34) implies the first claim.

Similarly, we can compute that

$$\langle \nabla J_T(L), E\rangle = \langle \mathbb{E}\left[\nabla\varepsilon(L, \mathcal{Y}_T)\right], E\rangle$$
$$= 2\mathrm{tr}\left[(I \otimes L)\mathcal{R}(I \otimes E^\mathsf{T})\mathcal{A}_L^\mathsf{T} H^\mathsf{T} H \mathcal{A}_L\right] + \mathrm{tr}\left[(\mathcal{Q} + (I \otimes L)\mathcal{R}(I \otimes L^\mathsf{T}))\mathcal{N}_L[E]\right],$$

and thus

$$\langle \nabla\widehat{J}_T(L) - \nabla J_T(L), E\rangle = -2\mathrm{tr}\left[\mathcal{Z}_3(I \otimes E^\mathsf{T})\mathcal{A}_L^\mathsf{T} H^\mathsf{T} H \mathcal{A}_L\right]$$
$$+ 2\mathrm{tr}\left[(I \otimes L)(\mathcal{Z}_2 - \mathcal{R})(I \otimes E^\mathsf{T})\mathcal{A}_L^\mathsf{T} H^\mathsf{T} H \mathcal{A}_L\right]$$
$$+ \mathrm{tr}\left[(\mathcal{Z}_L - \mathbb{E}\left[\mathcal{Z}_L\right])\mathcal{N}_L[E]\right].$$

Thus, by cyclic permutation property of trace and (12) we obtain that

$$|\langle \nabla \widehat{J}_T(L) - \nabla J_T(L), E \rangle| \le \| \frac{1}{M} \sum_{i=1}^{M} S_L(E, \mathcal{Y}_T^i) - \mathbb{E}\left[ S_L(E, \mathcal{Y}_T^i) \right] \| \| H^\intercal H \|_* \qquad (35)$$

where $S_L(E, \mathcal{Y})$ is the symmetric part of the following random matrix

$$-2\mathcal{A}_L \vec{\xi} \vec{\omega}^\intercal (I \otimes E^\intercal) \mathcal{A}_L^\intercal + 2\mathcal{A}_L(I \otimes L)\vec{\omega} \vec{\omega}^\intercal (I \otimes E^\intercal)\mathcal{A}_L^\intercal + 2\mathcal{A}_L \mathcal{X}_L \mathcal{M}_L[E]^\intercal.$$

Next, we provide the following almost sure bounds for each term: first,

$$\| \mathcal{A}_L \mathcal{Z}_3 (I \otimes E^\intercal) \mathcal{A}_L^\intercal \| \le \| \mathcal{A}_L \vec{\xi} \| \, \| \mathcal{A}_L (I \otimes E)\vec{\omega} \|$$

$$\le \kappa_\xi \left[ \sum_{i=0}^{T} \| A_L^i \| \right] \kappa_\omega \| E \| \left[ \sum_{i=0}^{T} \| A_L^i \| \right]$$

$$\le \kappa_\xi \kappa_\omega \frac{C_L^2}{[1 - \sqrt{\rho(A_L)}]^2} \| E \|$$

where the last equality follows by Lemma 6; second, similarly

$$\| \mathcal{A}_L(I \otimes L)\vec{\omega} \vec{\omega}^\intercal (I \otimes E^\intercal) \mathcal{A}_L^\intercal \|$$

$$\le \| \mathcal{A}_L(I \otimes L)\vec{\omega} \| \, \| \mathcal{A}_L(I \otimes E)\vec{\omega} \|$$

$$\le \kappa_\omega^2 \| L \| \, \| E \| \left[ \sum_{i=0}^{T} \| A_L^i \| \right]^2$$

$$\le \kappa_\omega^2 \frac{C_L^2}{[1 - \sqrt{\rho(A_L)}]^2} \| L \| \, \| E \|;$$

and finally

$$\| \mathcal{A}_L \mathcal{X}_L \mathcal{M}_L[E]^\intercal \| \le \| \mathcal{A}_L \vec{\eta}_L \| \, \| \mathcal{M}_L[E] \vec{\eta}_L \|$$

$$\le \kappa_L^2 \left[ \sum_{i=0}^{T} \| A_L^i \| \right] \left[ \sum_{i=0}^{T} \| M_i[E] \| \right]$$

$$\le \kappa_L^2 \left[ \frac{C_L + 2 C_L^3 \rho(A_L)^{3/2}}{[1 - \sqrt{\rho(A_L)}]^3} \right] \| EH \|$$

where the last inequality follows by Lemma 6. Now, by combining the last three bounds we can claim that almost surely

$$\| S_L(E, \mathcal{Y}) \| \le \frac{\nu_L}{\| H^\intercal H \|_*} \| E \|.$$

This also implies that

$$\| \mathbb{E}\left[ S_L(E, \mathcal{Y})^2 \right] \| \le \mathbb{E}\left[ \| S_L(E, \mathcal{Y}) \|^2 \right] \le \frac{\nu_L^2}{\| H^\intercal H \|_*^2} \| E \|^2.$$

Therefore, by Lemma 9 we obtain that

$$\mathbb{P}\left[ \| \frac{1}{M} \sum_{i=1}^{M} S_L(E, \mathcal{Y}_T^i) - \mathbb{E}\left[ S_L(E, \mathcal{Y}_T^i) \right] \| \ge t \right]$$

$$\le 2n \exp \left[ \frac{-M \| H^\intercal H \|_*^2 t^2/2}{\nu_L^2 \| E \|^2 + 2\nu_L \| H^\intercal H \|_* \| E \| t/3} \right]$$

Thus, by substituting $t$ with $t\|E\| / \| H^\intercal H \|_*$ and applying this bound to (35) we obtain that

$$\mathbb{P}\left[ |\langle \nabla \widehat{J}_T(L) - \nabla J_T(L), E \rangle| \le t\|E\| \right] \ge 1 - 2n \exp \left[ \frac{-M t^2/2}{\nu_L^2 + 2\nu_L t/3} \right]$$

Finally, choosing $E = \nabla \widehat{J}_T(L) - \nabla J_T(L)$ proves the second claim. $\qquad \square$

### F.4 Proof of the Proposition 5: Truncation error bound

We provide the proof for a detailed version of Proposition 5:

**Proposition 5'** (Truncation Error Bound). *Suppose $m_0 = 0$, then under Assumption 4 we have*

$$|J(L) - J_T(L)| \leq \bar{\xi}_L \frac{\rho(A_L)^{T+1}}{1 - \rho(A_L)},$$

*and*

$$\|\nabla J(L) - \nabla J_T(L)\| \leq \bar{\gamma}_L \frac{\sqrt{\rho(A_L)}^{T+1}}{[1 - \rho(A_L)]^2}$$

*where*

$$\bar{\xi}_L := \left[ \kappa_\xi^2 + (\kappa_\xi^2 + \kappa_\omega^2 \|L\|^2)C_L^2 \right] \|H^\intercal H\|_* C_L^2,$$

$$\bar{\gamma}_L := 2 \left[ \kappa_\xi^2 + C_L^2(\kappa_\xi^2 + \kappa_\omega^2 \|L\|^2) \right] C_L^2 \|H\| \|H^\intercal H\|_*$$
$$+ 2\kappa_\omega^2(\kappa_\xi^2 + \kappa_\omega^2 \|L\|^2)\|L\| \|H\|\|H^\intercal H\|_* \left( C_L + 2C_L^3 \rho(A_L)^{3/2} \right) C_L^3 \sqrt{\rho(A_L)}^{T+1}.$$

*Proof of Proposition 5.* For the purpose of this proof, we denote the same matrices by $\mathcal{A}_{L,T}$ and $\mathcal{M}_{L,T}[E]$ in order to emphasize on length $T$. Recall that

$$J_T(L) = \mathbb{E}\left[\varepsilon(L, \mathcal{Y})\right] = \mathrm{tr}\left[\mathbb{E}\left[\mathcal{X}_L\right] \mathcal{A}_{L,T}^\intercal H^\intercal H \mathcal{A}_{L,T}\right],$$

where $\mathbb{E}\left[\mathcal{X}_L\right] = \mathcal{Q} + (I \otimes L)\mathcal{R}(I \otimes L^\intercal)$, which implies

$$J_T(L) = \mathrm{tr}\left[[I \otimes (Q + LRL^\intercal)]\mathcal{A}_{L,T-1}^\intercal H^\intercal H \mathcal{A}_{L,T-1}\right] + \mathrm{tr}\left[P_0(A_L^\intercal)^T H^\intercal H A_L^T\right],$$

On the other hand,

$$J(L) = \lim_{t \to \infty} \mathrm{tr}\left[[I \otimes (Q + LRL^\intercal)]\mathcal{A}_{L,t}^\intercal H^\intercal H \mathcal{A}_{L,t}\right],$$

and thus

$$J(L) - J_T(L) = -\,\mathrm{tr}\left[P_0(A_L^\intercal)^T H^\intercal H A_L^T\right]$$
$$+ \lim_{t \to \infty} \mathrm{tr}\left[[I \otimes (Q + LRL^\intercal)][I \otimes (A_L^\intercal)^T]\mathcal{A}_{L,t}^\intercal H^\intercal H \mathcal{A}_{L,t}[I \otimes A_L^T]\right]$$
$$= -\,\mathrm{tr}\left[A_L^T P_0(A_L^\intercal)^T H^\intercal H\right]$$
$$+ \lim_{t \to \infty} \mathrm{tr}\left[[I \otimes A_L^T(Q + LRL^\intercal)(A_L^\intercal)^T]\mathcal{A}_{L,t}^\intercal H^\intercal H \mathcal{A}_{L,t}\right].$$

Therefore, by the properties of trace and Lemma 6 we obtain that

$$|J(L) - J_T(L)| \leq \|P_0\| \|A_L^T\|^2 \,\mathrm{tr}\,[H^\intercal H]$$
$$+ \|Q + LRL^\intercal\| \|(A_L)^T\|^2 \lim_{t \to \infty} \mathrm{tr}\left[\mathcal{A}_{L,t}^\intercal H^\intercal H \mathcal{A}_{L,t}\right]$$
$$\leq \|P_0\| C_L^2 \rho(A_L)^{T+1} \|H^\intercal H\|_*$$
$$+ (\|Q\| + \|R\| \|L\|^2)C_L^2 \rho(A_L)^{T+1} \frac{C_L^2 \|H^\intercal H\|_*}{1 - \rho(A_L)},$$

where the last line follows by Lemma 8. So,

$$|J(L) - J_T(L)| \leq \left[\|P_0\| + \frac{(\|Q\| + \|R\| \|L\|^2)C_L^2}{1 - \rho(A_L)}\right] \|H^\intercal H\|_* C_L^2 \rho(A_L)^{T+1}$$

This, together with Assumption 4 imply the first claim.

Next, for simplicity we adopt the notation $\mathcal{A}_{L,\infty}$ to interpret the limit as $t \to \infty$, then similar to the proof of Proposition 4 we can compute that

$$\langle \nabla J(L) - \nabla J_T(L), E \rangle = -\,2\mathrm{tr}\left[M_T[E]P_0(A_L^\intercal)^T H^\intercal H\right]$$
$$+ \mathrm{tr}\left[[I \otimes A_L^T(Q + LRL^\intercal)(A_L^\intercal)^T]\mathcal{N}_{L,\infty}[E]\right]$$
$$+ 2\mathrm{tr}\left[[I \otimes M_T[E](Q + LRL^\intercal)(A_L^\intercal)^T]\mathcal{A}_{L,\infty}^\intercal H^\intercal H \mathcal{A}_{L,\infty}\right]$$
$$+ 2\mathrm{tr}\left[[I \otimes A_L^T ERL^\intercal(A_L^\intercal)^T]\mathcal{A}_{L,\infty}^\intercal H^\intercal H \mathcal{A}_{L,\infty}\right].$$

Therefore, using (12) and Lemma 6 we have the following bound

$$
\begin{aligned}
|\langle \nabla J(L) - \nabla J_T(L), E\rangle| \leq &2\|EH\|C_L^2(T+1)\rho(A_L)^{T+1}\|P_0\|\|H^\intercal H\|_* \\
&+ \|Q + LRL^\intercal\|C_L^2\rho(A_L)^{T+1}\|\mathcal{N}_{L,\infty}[E]\|_* \\
&+ 2\|Q + LRL^\intercal\|\|EH\|C_L^2(T+1)\rho(A_L)^{T+1}\|\mathcal{A}_{L,\infty}^\intercal H^\intercal H\mathcal{A}_{L,\infty}\|_* \\
&+ 2\|ERL^\intercal\|C_L^2\rho(A_L)^{T+1}\|\mathcal{A}_{L,\infty}^\intercal H^\intercal H\mathcal{A}_{L,\infty}\|_*
\end{aligned}
$$

which by Lemma 8 is bounded as follows

$$
\begin{aligned}
|\langle \nabla J(L) - \nabla J_T(L), E/\|E\|\rangle| \leq &2\|P_0\|\|H\|\|H^\intercal H\|_* C_L^2(T+1)\rho(A_L)^{T+1} \\
&+ \|Q + LRL^\intercal\|\|H\|\,\|H^\intercal H\|_* \frac{\left[2C_L^3 + 4C_L^5\rho(A_L)^{3/2}\right]\rho(A_L)^{T+1}}{[1-\rho(A_L)]^2} \\
&+ 2\|Q + LRL^\intercal\|\|H\|\|H^\intercal H\|_* \frac{C_L^4(T+1)\rho(A_L)^{T+1}}{1-\rho(A_L)} \\
&+ 2\|R\|\|L\|\|H^\intercal H\|_* \frac{C_L^4\rho(A_L)^{T+1}}{1-\rho(A_L)}.
\end{aligned}
$$

Finally, choosing $E = \nabla J(L) - \nabla J_T(L)$ together with Assumption 4 implies

$$
\begin{aligned}
\|\nabla J(L) - \nabla J_T(L)\| \leq &2\left[\frac{\kappa_\xi^2 + C_L^2(\kappa_\xi^2 + \kappa_\omega^2\|L\|^2)}{1-\rho(A_L)}\right]C_L^2\|H\|\,\|H^\intercal H\|_*(T+1)\rho(A_L)^{T+1} \\
&+ 2\left[\frac{\kappa_\omega^2(\kappa_\xi^2 + \kappa_\omega^2\|L\|^2)\|L\|\,\|H\|\left(C_L + 2C_L^3\rho(A_L)^{3/2}\right)}{1-\rho(A_L)}\right]\|H^\intercal H\|_* \frac{C_L^3\rho(A_L)^{T+1}}{1-\rho(A_L)}.
\end{aligned}
$$

Finally, the second claim follows by the following simple facts:

$$
(T+1)\rho(A_L)^{T+1} \leq \frac{\sqrt{\rho(A_L)}^{T+1}}{1-\rho(A_L)}, \quad \forall T > 0,
$$

as $\max_{t \geq 0} t\rho^t = \frac{2}{e\ln 1/\rho} \leq \frac{1}{\ln 1/\rho} \leq \frac{1}{1-\rho}$ for any $\rho \in (0,1)$. This completes the proof. $\qquad\square$

### F.5 Complete version of Theorem 2: Sample complexity bounds for the stochastic oracle

The following is a detailed version of Theorem 2:

**Theorem 2'.** *Suppose $m_0 = 0_n$ and Assumption 4 holds for a data-set $\{\mathcal{Y}_T^i\}_{i=1}^M$. Define $\nabla \widehat{J}_T(L) := \frac{1}{M}\sum_{i=1}^M \nabla\varepsilon(L, \mathcal{Y}_T^i)$, where $\nabla_L\varepsilon(L, \mathcal{Y})$ is obtained in Lemma 3. Consider $\mathcal{S}_\alpha$ for some $\alpha > 0$ and any $s, s_0 > 0$ and $\tau \in (0,1)$. Suppose the trajectory length*

$$
T \geq \ln\left(\frac{\bar{\gamma}_\alpha\sqrt{\min(n,m)}}{s_0}\right) \Big/ \ln\left(\frac{1}{\sqrt{\rho_\alpha}}\right)
$$

*and the batch size*

$$
M \geq \left[2\left(\frac{\nu_\alpha\sqrt{\min(n,m)}}{s\,s_0\,/\tau}\right)^2 + \frac{4}{3}\left(\frac{\nu_\alpha\sqrt{\min(n,m)}}{s\,s_0\,/\tau}\right)\right]\ln(2n/\delta),
$$

*where*

$$
\begin{aligned}
\bar{\gamma}_\alpha := \ &2\left[\kappa_\xi^2 + C_\alpha^2(\kappa_\xi^2 + \kappa_\omega^2 D_\alpha^2)\right]C_\alpha^2\|H\|\,\|H^\intercal H\|_* \\
&+ 2\kappa_\omega^2(\kappa_\xi^2 + \kappa_\omega^2 D_\alpha^2)D_\alpha\|H\|\|H^\intercal H\|_*\left(C_\alpha + 2C_\alpha^3\rho_\alpha^{3/2}\right)C_\alpha^3\sqrt{\rho_\alpha}^{T+1},
\end{aligned}
$$

$$
\nu_\alpha := \frac{2(\kappa_\xi + D_\alpha\kappa_\omega)\kappa_\omega C_\alpha^2 + \left[C_\alpha + 2C_\alpha^3\rho_\alpha^{3/2}\right]\|H\|(\kappa_\xi + D_\alpha\kappa_\omega)^2}{[1-\sqrt{\rho_\alpha}]^3/\|H^\intercal H\|_*},
$$

*with $\rho_\alpha$, $C_\alpha$ and $D_\alpha$ defined in Lemma 6. Then, with probability no less than $1-\delta$, Assumption 3 holds.*

### F.6 Additional concentration bound results

Combining the truncation bound in Proposition 5 with concentration bounds in Proposition 4 we can provide probabilistic bounds on the "estimated cost" $\widehat{J}_T(L)$ and the "estimated gradient" $\nabla\widehat{J}_T(L)$. The result involves the bound for the Frobenius norm of the error with probabilities independent of $T$.

Theorem 2, can be viewed as a simplified application of Theorem 5 to characterize the required minimum trajectory length and minimum batch so that the approximate gradient satisfies Assumption 3, with a specific $s$ and $s_0$.

**Theorem 5.** *Suppose Assumption 4 holds. For any $s > 0$ and $L \in \mathcal{S}_\alpha$, if*

$$M \geq \left[2\left[\frac{\nu_L\sqrt{\min(n,m)}}{s\,\|\nabla J(L)\|_F}\right]^2 + \frac{4}{3}\left[\frac{\nu_L\sqrt{\min(n,m)}}{s\,\|\nabla J(L)\|_F}\right]\right]\ln(2n/\delta),$$

*then with probability no less than $1 - \delta$,*

$$\|\nabla\widehat{J}_T(L) - \nabla J(L)\|_F \leq s\|\nabla J(L)\|_F + \bar{\gamma}_L\sqrt{\min(n,m)}\sqrt{\rho(A_L)}^{T+1},$$

*with $\nu_L$ and $\bar{\gamma}_L$ defined in Proposition 4 and Proposition 5, respectively.*

*Proof of Theorem 5.* Recall that for any $L \in \mathcal{S}_\alpha$ for some $\alpha > 0$ we have

$$\|\nabla\widehat{J}_T(L) - \nabla J(L)\| \leq \|\nabla\widehat{J}_T(L) - \nabla J_T(L)\| + \|\nabla J_T(L) - \nabla J(L)\|.$$

Thus, by Proposition 4 with $s$ replaced by $s\|\nabla J(L)\|_F/\sqrt{\min(n,m)}$ and applying Proposition 5 to the second term, we obtain that with probability at least $1 - \delta$:

$$\|\nabla\widehat{J}_T(L) - \nabla J(L)\| \leq \frac{s\|\nabla J(L)\|_F}{\sqrt{\min(n,m)}} + \bar{\gamma}_L\frac{\sqrt{\rho(A_L)}^{T+1}}{[1 - \rho(A_L)]^2},$$

where

$$\delta \geq 2n\exp\left[\frac{-Ms^2/2}{\left[\frac{\nu_L\sqrt{\min(n,m)}}{\|\nabla J(L)\|_F}\right]^2 + 2\left[\frac{\nu_L\sqrt{\min(n,m)}}{\|\nabla J(L)\|_F}\right]s/3}\right].$$

Noticing $\|\nabla\widehat{J}_T(L) - \nabla J(L)\|_F \leq \sqrt{\min(n,m)}\|\nabla\widehat{J}_T(L) - \nabla J(L)\|$ and rearranging terms will complete the proof. $\quad\square$

One can also provide the analogous concentration error bounds where the probabilities are independent of the system dimension $n$.

**Proposition 7** (Concentration independent of system dimension $n$). *Under the same hypothesis, we have*

$$\mathbb{P}\left[|\widehat{J}_T(L) - J_T(L)| \leq s\right] \geq 1 - 2T\exp\left[\frac{-Ms^2/2}{\bar{\mu}_L^2 T^2 + 2\bar{\mu}_L Ts/3}\right],$$

*and*

$$\mathbb{P}\left[\|\nabla\widehat{J}_T(L) - \nabla J_T(L)\| \leq s\right] \geq 1 - 2T\exp\left[\frac{-Ms^2/2}{\bar{\nu}_L^2 T^2 + 2\bar{\nu}_L Ts/3}\right]$$

$$- 2T\exp\left[\frac{-Ms^2/2}{\kappa_\omega^2\bar{\mu}_L^2 T^2 + 2\kappa_\omega\bar{\mu}_L Ts/3}\right]$$

*where*

$$\bar{\mu}_L := \frac{C_L^2\|H^\intercal H\|_*}{1 - \rho(A_L)}\kappa_L^2$$

$$\bar{\nu}_L := \frac{\left[2C_L + 4C_L^3\rho(A_L)^{3/2}\right]\|H\|\,\|H^\intercal H\|_*}{[1 - \rho(A_L)]^2}\kappa_L^2.$$

*Proof of Proposition 7.* Similar to the previous proof, we have

$$\widehat{J}_T(L) - J_T(L) = \operatorname{tr}\left[(\mathcal{Z}_L - \mathbb{E}\left[\mathcal{Z}_L\right])\mathcal{A}_L^\intercal H^\intercal H\mathcal{A}_L\right],$$

and thus, by (12) we obtain

$$|\widehat{J}_T(L) - J_T(L)| \leq \|(\mathcal{Z}_L - \mathbb{E}\left[\mathcal{Z}_L\right])\|\,\|\mathcal{A}_L^\intercal H^\intercal H\mathcal{A}_L\|_*. \tag{36}$$

Next, we consider the symmetric random matrix $(\mathcal{Z}_L - \mathbb{E}\left[\mathcal{Z}_L\right])$ and recall that $\|\xi(t) - L\omega(t)\| \leq \kappa_L$ almost surely; thus

$$\|\mathcal{X}_L\| = \|\vec{\eta}_L\|^2 \leq \kappa_L^2 T^2.$$

It then follows that

$$\left\|\mathbb{E}\left[\mathcal{X}_L^2\right]\right\| \leq \mathbb{E}\left[\|\mathcal{X}_L\|^2\right] \leq \kappa_L^4 T^4.$$

Therefore, by Lemma 9 we obtain that

$$\mathbb{P}\left[\|(\mathcal{Z}_L - \mathbb{E}\left[\mathcal{Z}_L\right])\| \geq t\right] \leq 2T\exp\left[\frac{-Mt^2/2}{\kappa_L^4 T^4 + 2\kappa_L^2 T^2 t/3}\right] \tag{37}$$

Substituting $t$ with $t/\|\mathcal{A}_L^\intercal H^\intercal H\mathcal{A}_L\|_*$ together with (36) implies the first claim because by Lemma 8 $\kappa_L^2 T^2\|\mathcal{A}_L^\intercal H^\intercal H\mathcal{A}_L\|_* \leq \bar{\mu}_L T$.

Again, similar to (35) in the previous proof, by (12) we obtain that

$$|\langle\nabla\widehat{J}_T(L) - \nabla J_T(L), E\rangle| \leq \|\frac{1}{M}\sum_{i=1}^{M}S_L(E, \mathcal{Y}_T^i) - \mathbb{E}\left[S_L(E, \mathcal{Y}_T^i)\right]\|\,\|\mathcal{A}_L^\intercal H^\intercal H\mathcal{A}\|_*$$

$$+ \|\frac{1}{M}\sum_{i=1}^{M}\mathcal{X}_L(\mathcal{Y}^i) - \mathbb{E}\left[\mathcal{X}_L(\mathcal{Y}^i)\right]\|\,\|\mathcal{N}_L[E]\|_* \tag{38}$$

where $S_L(E, \mathcal{Y})$ is the symmetric part of the following random matrix

$$-2\vec{\xi}\vec{\omega}^\intercal(I \otimes E^\intercal) + 2(I \otimes L)\vec{\omega}\vec{\omega}^\intercal(I \otimes E^\intercal).$$

So, we claim that almost surely

$$\|S_L(E, \mathcal{Y})\| \leq 2\|(I \otimes E)\vec{\omega}\|(\|\vec{\xi}\| + \|(I \otimes L)\vec{\omega}\|)$$
$$\leq 2\|E\|\kappa_\omega T(\kappa_\xi T + \|L\|\kappa_\omega T)$$
$$= \kappa_L\kappa_\omega T^2\|E\|,$$

and thus

$$\|\mathbb{E}\left[S_L(E, \mathcal{Y})^2\right]\| \leq \mathbb{E}\left[\|S_L(E, \mathcal{Y})\|^2\right] \leq \kappa_L^2\kappa_\omega^2 T^4\|E\|^2.$$

Therefore, by Lemma 9 we obtain that

$$\mathbb{P}\left[\|\frac{1}{M}\sum_{i=1}^{M}S_L(E, \mathcal{Y}_T^i) - \mathbb{E}\left[S_L(E, \mathcal{Y}_T^i)\right]\| \geq t\right] \leq 2T\exp\left[\frac{-Mt^2/2}{\kappa_\omega^2\kappa_L^2 T^4\|E\|^2 + 2\kappa_L\kappa_\omega T^2\|E\|t/3}\right]$$

Substituting $t$ with $t/\|\mathcal{A}_L^\intercal H^\intercal H\mathcal{A}_L\|_*$ implies that

$$\mathbb{P}\left[\|\frac{1}{M}\sum_{i=1}^{M}S_L(E, \mathcal{Y}_T^i) - \mathbb{E}\left[S_L(E, \mathcal{Y}_T^i)\right]\|\,\|\mathcal{A}_L^\intercal H^\intercal H\mathcal{A}_L\|_* \geq t\right]$$

$$\leq 2T\exp\left[\frac{-Mt^2/2}{\kappa_\omega^2\bar{\mu}_L^2 T^2\|E\|^2 + 2\kappa_\omega\bar{\mu}_L T\|E\|t/3}\right]$$

because by Lemma 8 we have $\kappa_L^2 T^2\|\mathcal{A}_L^\intercal H^\intercal H\mathcal{A}_L\|_* \leq \bar{\mu}_L T$.

Next, by substituting $t$ with $t/\|\mathcal{N}_L[E]\|_*$ in (37) we have

$$\mathbb{P}\left[\|\mathcal{Z}_L - \mathbb{E}\left[\mathcal{Z}_L\right]\|\,\|\mathcal{N}_L[E]\|_* \geq t\right] \leq 2T\exp\left[\frac{-Mt^2/2}{\bar{\nu}_L^2 T^2\|E\|^2 + 2\bar{\nu}_L T\|E\|t/3}\right]$$

because Lemma 8 implies that $\kappa_L^2 T^2 \|\mathcal{N}_L[E]\|_* \leq \bar{\nu}_L T \|E\|$. Thus, by combining the last two inequalities and using the union bound for (38) we obtain that

$$\mathbb{P}\left[|\langle \nabla \widehat{J}_T(L) - \nabla J_T(L), E \rangle| \geq t\right] \leq 2T \exp\left[\frac{-Mt^2/2}{\bar{\nu}_L^2 T^2 \|E\|^2 + 2\bar{\nu}_L T \|E\| t/3}\right]$$
$$+ 2T \exp\left[\frac{-Mt^2/2}{\kappa_\omega^2 \bar{\mu}_L^2 T^2 \|E\|^2 + 2\kappa_\omega \bar{\mu}_L T \|E\| t/3}\right]$$

Finally, substituting $t$ with $t\|E\|$ and choosing $E = \nabla \widehat{J}_T(L) - \nabla J_T(L)$ proves the second claim. $\square$

*Remark* 10. We obtained a better bound for truncation of the gradient as

$$\|\nabla J(L) - \nabla J_T(L)\| \leq \bar{\gamma}_1(L) \frac{\sqrt{\rho(A_L)}^{T+1}}{[1 - \rho(A_L)] \ln(1/\rho(A_L))} + \bar{\gamma}_2(L) \frac{\rho(A_L)^{T+1}}{[1 - \rho(A_L)]^2},$$

which has been simplified for clarity of the presentation.

## Appendix G   Numerical Results

Herein, we showcase the application of the developed theory for improving the estimation policy for an LTI system. Specifically, we consider an undamped mass-spring system with known parameters $(A, H)$ with $n = 2$ and $m = 1$. In the hindsight, we consider a variance of $0.1$ for each state dynamic noise, a state covariance of $0.05$ and a variance of $0.1$ for the observation noise. Assuming a trajectory of length $T$ at every iteration, the approximate gradient is obtained as in Lemma 3, only requiring an output data sequence collected from the system in (1). Then, the progress of policy updates using the SGD algorithm for different values of trajectory length $T$ and batch size $M$ are depicted in Figure 1 where each figure shows *average progress* over 50 rounds of simulation. The figure demonstrates a linear convergence outside of a neighborhood of global optimum that depends on the bias term in the approximate gradient (due to truncated data trajectories). The rate then drops when the policy iterates enter into this neighborhood which is expected as every update only relies on a *biased gradient*—in contrast to the linear convergence established for deterministic GD (to the exact optimum) using the true gradient.

Specifically, recall that our convergence guarantee to a small neighborhood around the optimal value is due to the finite-length of the data trajectories. The region can be made arbitrary small by choosing larger trajectory length. In particular, to achieve $\varepsilon$ error, we only require the length $T \geq O(\ln(1/\varepsilon))$—see Theorem 3. Also, Fig 1(d) is illustrating that optimality gap at the final iteration (i.e. the radius of the small neighborhood around optimality) which is decaying linearly as a function of trajectory length $T \leq 50$—until the variance error dominates beyond $T = 50$. It is clear that, increasing the batch size $M$ will allow further decrease of this optimality gap beyond $T = 50$.

The code for regenerating these results is available online at this GitHub repository [55].

## Appendix H   Nomenclature

**Constant Values**

| | |
|---|---|
| $\alpha$ | Scalar value of the cost |
| $\bar{\gamma}_\alpha$ | Uniform constant for gradient estimate—see Theorem 2 |
| $\bar{\gamma}_L$ | Constant for bounding bias of gradient estimate—see Proposition 5 |
| $\delta$ | Failure probability |
| $\eta$ | Step-size of SGD algorithm |
| $\gamma$ | Positive constant smaller than one—see Proposition 2 |
| $\kappa_\omega$ | Scalar upperbound on 2-norm of measurement noise |
| $\kappa_\xi$ | Scalar upperbound on 2-norm of dynamic noise |

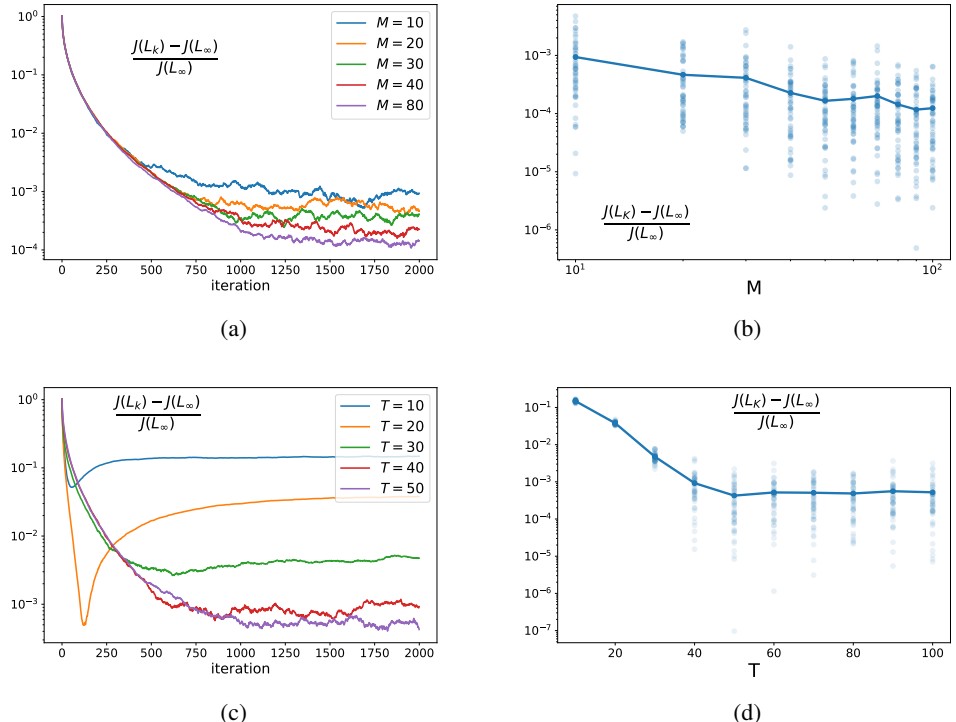

Figure 1: Simulation result of the SGD algorithm to learn the steady-state Kalman gain for the mass-spring example. (a) The optimality gap as a function of iterations $k$ for different value of batch-size $M$ averaged over 50 simulations; (b) The optimality gap at final iteration as a function of batch-size $M$ for all 50 simulations; (c) The optimality gap as a function of iterations $k$ for different value trajectory length $T$. The figure depicts the linear decay of the optimality gap, with respect to the iteration $k$, before the iterate enters the small neighborhood of optimality where the direction of the oracle gradient is not informative anymore. The neighborhood shrinks as $T$ increases; (d) The optimality gap at final iteration as a function of trajectory length $T$. The gap is decaying linearly as a function of trajectory length $T$ until the variance error dominates beyond $T = 50$.

| | |
|---|---|
| $\kappa_L$ | Combined Scalar upperbound on 2-norm of noise |
| $\nu_\alpha$ | Uniform constant for gradient estimate—see Theorem 2 |
| $\nu_L$ | Combined matrix variance statistics of gradient estimate—see Proposition 4 |
| $k$ | Iteration step of SGD algorithm |
| $M$ | Batch-size of SGD algorithm |
| $m$ | Dimension of observation vector |
| $n$ | Dimension of state vector |
| $s$ | Variance coefficient term of oracle model—see Assumption 3 |
| $s_0$ | Bias term of oracle model—see Assumption 3 |
| $T$ | Length of output data trajectory |
| $t$ | Time step of the system |

**Other symbols**

| | |
|---|---|
| $\|\cdot\|$ | 2-norm of a matrix |
| $\|\cdot\|_F$ | Frobenius-norm of a matrix |
| $\mathcal{L}$ | Linear operator describing the filtering policy that maps measurements to state estimation |
| $\mathcal{L}^\dagger$ | Adjoint of the operator $\mathcal{L}$ |

$\rho(\cdot)$       Spectral radius of a matrix

$\rho_\alpha$       Uniform upperbound of spectral radius of closed-loop system over $\mathcal{S}_\alpha$–see Lemma 6

$c_1, c_2, c_3$   Uniform positive constants related to the cost function on $\mathcal{S}_\alpha$–see Lemma 2

$C_\alpha$       Uniform upperbound of constant coefficient over $\mathcal{S}_\alpha$–see Lemma 6

$C_L$       Constant coefficient of upperbound in Lemma 6

$D_\alpha$       Uniform upperbound of 2-norm of policy $L$ over $\mathcal{S}_\alpha$–see Lemma 6

**Problem Parameters**

$\mathcal{A}_L$       Concatenation of closed-loop dynamics matrices—see Proposition 3

$A$       Dynamics matrix

$H$       Observation matrix

$m_0$       Initial state mean vector

$P$       Error covariance matrix

$P_0$       Initial state covariance matrix

$Q$       State noise covariance matrix

$R$       Output (measurement) noise covariance matrix

**System Quantities**

$\eta$       Combined noise vector

$\hat{x}$       Estimation of state vector

$\hat{y}$       Estimation of output vector

$\mathcal{C}_\tau$       $s_0/\tau$-neighborhood of the globally optimal policy $L^*$

$\mathcal{M}_L$       Concatenated matrix regarding the differential of $\varepsilon(L, \mathcal{Y}_T)$—see Proposition 3

$\mathcal{N}_L$       Concatenated matrix regarding the differential of $\varepsilon(L, \mathcal{Y}_T)$—see Proposition 3

$\mathcal{S}$       Set of Schur stabilizing policies

$\mathcal{S}_\alpha$       $\alpha$-sublevel set of $J$ contained in $\mathcal{S}$

$\mathcal{U}$       Trajectory of adjoint input vector

$\mathcal{X}_L$       Concatenated matrix regarding the differential of $\varepsilon(L, \mathcal{Y}_T)$—see Proposition 3

$\mathcal{Y}$       Trajectory of output vectors

$\nabla\widehat{J}$       Estimation of gradient of cost function

$\nabla J$       Gradient of cost function

$\omega$       Output (measurement) noise vector

$\partial\mathcal{S}$       Boundary of the set of Schur stabilizing policies

$\varepsilon(L, \mathcal{Y}_T)$   Squared-norm of the estimation error vector of policy $L$ for trajectory $\mathcal{Y}_T$

$\vec{\eta}$       Concatenated combined noise vector

$\vec{\omega}$       Concatenated measurement noise vector

$\vec{\xi}$       Concatenated state noise vector

$\widehat{J}_T$       Estimation of the cost function on length $T$ trajectory

$\xi$       State (dynamics) noise vector

$e_T(L)$       Estimation error vector of policy $L$ for trajectory $\mathcal{Y}_T$

$J$       Cost function

$J_T$       Cost function on length $T$ trajectory

$L$       Filtering policy matrix

$u$       Adjoint input vector

$X$       Cost matrix

$x$       State vector

$X_T$      Cost matrix on length $T$ trajectory

$Y$      Auxiliary cost matrix

$y$      Output (measurement) vector

$z$      Adjoint state vector

