# OpenReview forum: "Data-driven Optimal Filtering for Linear Systems with Unknown Noise Covariances"
_NeurIPS.cc/2023/Conference — NeurIPS 2023 poster_

### Official Review · Reviewer_4EhD · 2023-07-01

**Soundness:** 3 good
**Presentation:** 2 fair
**Contribution:** 3 good
**Rating:** 8
**Confidence:** 3

**Summary:**

The paper builds on the duality between estimation and control, in order to develop an online data-driven method for MSE-optimal filtering of linear systems with linear observations. Process and observation noise covariances are considered unknown, but stochastic states are assumed to be bounded.  The paper proposes using SGD in the space of steady-state  stabilizing gains, claims asymptotic convergence to the optimal gain, and provides an asymptotic probabilistic bound on the deviation from optimal error.

**Strengths:**

Although the use of online optimization for controlling linear-quadratic settings is not new (e.g. [r1]), exploiting the duality between control and filtering problems is novel and interesting in this context. The concentration and error bounds are not trivial and useful, as concentration bound is non-asymptotic in the series length T.
Proof of SGD convergence and error bounds are novel as well, even if derived under very strong assumptions.

[r1] Cohen A. et al., "Online Linear Quadratic Control", 2018



**Weaknesses:**

Overall, I believe the authors did their best that the paper will be well-organized and clear (as it can be for a rather technical manuscript). The explanations and given outlines before each section are indeed helpful.  However, it is still very hard to follow the assumptions and constant definitions, and some of the conclusions. Writing becomes very laconic at some crucial points (e.g. Thm. 2, Remark 7).

Particularly, Thm.2 and it's proof are not clear hence it is hard to get convinced in their soundness.  Furthermore, In the introduction it is stated that convergence is guaranteed from every initial policy, but according to Theorem 2 the policy cannot enter (or start in) a class of policies where gradient is smaller than some constant. I didn't find any discussion about when trajectories enter this region. My impression is that this is a good paper, and I might be missing something. I will be willing to raise my score when given a more detailed proof and this clarification.


**Questions:**

I ask for a more detailed proof for Thm. 2 (see above).

In addition, a summary of all assumptions, results and notations can be very useful.



Minor typos:

l. 249 't' should be replaced by $\gamma$.

l. 276 I think i should go between 1 and T-1.

l. 309 quite.


**Limitations:**

The online method uses a surrogate loss using the observations, this is a reasonable choice due to lack of ground-truth states and the observability, which is a strong assumption. The paper should discuss the implications of using this loss with more general (i.e. non observable) systems.

---

> ### Author Rebuttal · Authors · 2023-08-09
>
> **Q**: *... However, it is still very hard to follow the assumptions...*
>
> **A**: Please see the Response to All for clarification on the technical results and the changes we made for the revision.
>
> **Q**: *...Particularly, Thm.2 and it's proof are not clear ...*
>
> **A**: We provide the following observation that clearly states the implications of Theorem 2 followed by its detailed proof:
>
> *Observation 1*. Combining Theorem 2 and the PL property (11a) results in a sample complexity for our
> algorithm. In particular, it follows that  $J(L_k)-J(L^*)\leq \varepsilon$, if $s_0\leq \frac{\sqrt{c_1(\alpha) \varepsilon}}{4}$, $s\leq \frac{1}{4}$, and the number of steps
> $k>\ln(\frac{\varepsilon}{\alpha})/\ln(1-\frac{c_1(\alpha)}{18\ell(\alpha)}))$.
>
> While the original version of Theorem 2 is stated for any choice of $\epsilon \in (0,1)$, for simplicity we present its proof for the special case of $\epsilon = 1/2$:
>
> *Proof of Theorem 2.* The first step of the proof is to show that the
> assumption of the Proposition 2 is satisfied for
> $E=\nabla \widehat J(L)$ for all
> $L \in \mathcal{S}\_\alpha \setminus \mathcal{C}\_{\gamma/2}$. This is
> true because $$\begin{aligned}
>     \|\nabla \widehat J(L) - \nabla J(L)\| \leq s\|\nabla J(L)\| + s_0 \\
>     \leq s\|\nabla J(L)\| + \frac{\gamma}{2} \|\nabla J(L)\|\\
>     \leq \gamma \|\nabla J(L)\|
> \end{aligned}$$ where the first inequality follows from Assumption 3,
> the second inequality follows from $L \not \in C_{\gamma/2}$ (i.e.
> $s_0\|\nabla J(L)\|\geq \gamma/2$), and the last step follows from the
> assumption $s\leq \gamma/2$.\
> The rest of the proof relies on repeated application of Proposition 2.
> In particular, starting from
> $L_0 \in \mathcal{S}\_\alpha \setminus \mathcal{C}\_{\gamma/2}$, the
> application of Proposition 2 implies that
> $L_1\coloneqq L_0 - \bar\eta \nabla\widehat J(L_0)$ remains in the same
> sublevel set, i.e., $L_1 \in \mathcal{S}\_{\alpha}$, and we obtain the
> following linear decay of the cost value:
> $$J(L_1) -J(L^*) \leq \left[1-c_1(\alpha)\bar\eta{(1 - \gamma)}/2\right] [J(L_0)- J(L^*)].$$
> Now, if $L_1 \in \mathcal{C}\_{\gamma/2}$ then we stop; otherwise
> $L_1 \in \mathcal{S}\_\alpha \setminus \mathcal{C}\_{\gamma/2}$ and we can
> repeat the above process to arrive at
> $$J(L_2) -J(L^*) \leq \left[1-c_1(\alpha)\bar\eta{(1 - \gamma)}/2\right]^2 [J(L_0)- J(L^*)].$$
> Repeating the process generates a sequence of policies
> $L_0, L_1, L_2 ...$ with a combined linear decay of
> $$J(L_k) -J(L^*) \leq \left[1-c_1(\alpha)\bar\eta{(1 - \gamma)}/2\right]^k [J(L_0)- J(L^*)],$$
> unless at some iteration $j$, we arrive at a policy $L_j$ such that
> $L_j \in \mathcal{C}\_{\gamma/2}$. This completes the proof. ◻
>
>
>
> **Q**: *...convergence is guaranteed from every initial policy, but...I might be missing something...*
>
> **A**:
> We think the missing point here is the PL property of the cost that seems to understate the result of Theorem 2. As shown in Lemma 2 (and discussed in Remark 3), the cost maintain PL property on each sublevel set. In particular (11a) implies that on each $\mathcal S_\alpha$ , $\|\nabla J (L)\|\_F$ characterizes the optimality gap $J(L) - J(L^*)$ by:
> $$c_1(\alpha) [J(L) - J(L^*)] \leq  \|\nabla J(L)\|\_F^2$$
> for some constant $c_1(\alpha)$. This implies that if we have arrived at a candidate policy $L_k$ for which the gradient is small, then the optimality gap should be small (involving the constant $c_1(\alpha)$
> that is independent of $L_k$).
>
> This is the reason that in Theorem 2, it suffices to argue about the generated sequence $L_k$ to have a linear decay unless entering a small neighborhood of $L^*$ containing policies with small enough gradients (denoted by $\mathcal{C}\_\tau$). In particular, if for some $j<k$, we arrive at some policy $L_j \in \mathcal{C}\_\tau$, then by (11a) we can conclude that:
> $$J(L_j)-J(L^*) \leq \frac{1}{c_1(\alpha)} \|\nabla J(L_j)\|\_F^2  \leq \frac{s_0^2}{c_1(\alpha) \tau^2},$$
> which is directly controlled by the bias term $s_0$. This is the bound used also in the argument of Remark 7. We believe this is the missing point for causing the confusion about what happens when the trajectories enter this region.
>
> Additionally, as stated in the introduction, every (stabilizing) initial policy $L_0$ amounts to a finite value of $J(L_0)$ and thus lies in some
> sublevel set $S_\alpha$. So, starting from such $L_0$, Theorem 2
> guarantees linear decay of the optimality gap till the trajectory enters
> that small neighborhood. Finally, note that the radius of this
> neighborhood $\mathcal{C}_\tau$ is characterized by the bias term $s_0$
> which itself is exponentially decaying to zero in the trajectory length
> $T$.
>
>
> **Q:** *... a summary of all assumptions, results....*
>
> **A**: Assumption 1 sates that the linear system is detectable. This is the minimum requirement to make the estimation problem well-posed.
>
> Assumption 2 states that the linear system is observable. This stronger assumption is made in lieu
> of Assumption 1 to improve the clarity of the analysis with less system theoretic
> technicalities.
>
> Assumption 3 states an error bound on the gradient oracle. This is made in order to provide an independent analysis of the SGD algorithm
> for locally Lipschitz functions in presence of gradient bias. This assumption is later verified by Thm. 3.
>
> Assumption 4 assumes bounded process and observation noise. This is made to facilitate application of matrix concentration inequalities.
>
> Finally, we now have a complete nomenclature that is added to the supplementary materials.
>
> **Q**: *The online method uses a surrogate loss...*
>
> **A:** If the system is non-observable there is no guarantee for existence of a stabilizing policy unless the system is detectable (Assumption 1). In fact, detectability is a necessary and sufficient condition for well-posedness. While our analysis is presented for observable systems, we believe that the extension to detectable case is possible by adopting more system theoretic tools in Lemma 1.

---

> > ### Comment · Reviewer_4EhD · 2023-08-12
> >
> > I appreciate the time and efforts made to address my (and other reviewers) concerns. After carefully reading the response, I believe the missing points in Thm2's proof are now clear. I would recommend to clarify them in the text in order to improve readability for a broader audience. I will raise my score, accordingly.

---

### Official Review · Reviewer_yyZM · 2023-07-02

**Soundness:** 3 good
**Presentation:** 4 excellent
**Contribution:** 3 good
**Rating:** 6
**Confidence:** 4

**Summary:**

This submission examines the learning of the Kalman filter gain for linear systems with unknown covariance matrices using noisy output data. Similar to learning the linear quadratic regulators for unknown linear systems, the learning problem here is posed as a stochastic policy optimization problem which minimizes the expected output prediction error. Bridging together the learning of kalman gain and the optimal control gain, the paper provides an interesting convergence analysis of stochastic gradient descent for the policy optimization problem, and bias-variance error bounds of the learning problem by employing a set of tools from linear systems and stochastic geometry.

**Strengths:**

+) The learning of the Kalman gain as a stochastic policy optimization problem which is amendable to results and studies of learning linear quadratic regulators in the literature;
+) the dual of learning the Kalman gain and the optimal control gain in the data-driven setting;
+) biased gradients and stability constraints are strategically handled when learning the Kalman gain, along with the bias-variance error bounds.

**Weaknesses:**

-) The setup is not very practical as the system matrices are assumed perfectlt known but only the covariance matrices are unknown; even in e.g., the stated application aircraft wign dynamics where only approximate models are known, one would assume imperfect knowledge of both the system matrices and covariance matrices; or identifying them via data-driven method first.
-) For the learning problem described in lines 106-108, it is unclear whether the problem is well-posed in the sense that given these information (data), wheher one is able to learn the steady -state Kalman gain L_\infinity, or at least how large the horizon T shall be such that one will be able to have a unique L_\infty?
-) In (1), are uncorrelated random vectors enough for deriving the optimal Kalman filter without mutually independent random noise vectors? Also in the formulation of (5), what are random quantities and do we have any condition expectations or not?


**Questions:**

It would be great if the results can be compared with more related works on learning the Kalman filter under different settings; Zheng, Y. et al. Sample complexity of linear quadratic gaussian (LQG) control for output feedback systems. In Learning for dynamics and control (pp. 559-570). PMLR. Zhang, X. et al. Learning the Kalman filter with fine-grained sample complexity. arXiv preprint arXiv:2301.12624. In the behavioral theory literature, results are available for learning the Kalman filter for unknown linear systems by using data, although seemingly in addition to the noisy output data and also the pure state data, in e.g., Liu, W. et al. Learning Robust Data-based LQG Controllers from Noisy Data. arXiv preprint arXiv:2305.01417.

---

> ### Author Rebuttal · Authors · 2023-08-09
>
> We appreciate your constructive comments, clarifications, and reference to relevant literature that will indeed improve this manuscript.
>
> **Q:** *The setup is not very practical as the system matrices are assumed perfectly known but only the covariance matrices are unknown...*
>
> **A:** Please see our response to all.
>
> **Q:** *For the learning problem described in lines 106-108, it is unclear...*
>
> **A:** Please note that lines 106-108 are not intended as precise mathematical statement of the problem, but as a formal qualitative
> description: given independent realizations of output data with length $T$, our goal is to learn the steady-state Kalman gain. Our result
> (formerly Remark 7, now Theorem 1) shows that in order to learn the steady-state Kalman gain up to $\varepsilon$ accuracy, it is sufficient
> for the time horizon to satisfy $T\geq O(\ln(1/\varepsilon))$. The precise mathematical problem we intend to solve is the steady-state
> optimization problem (10).  The uniqueness of steady-state Kalman filter gain $L_\infty$ is guaranteed under Assumption 2.
>
> **Q:** *In (1), are uncorrelated random vectors enough for deriving the optimal Kalman filter...*
>
> **A:** Yes, uncorrelated random vectors is enough to establish that Kalman filter provides the best *linear* MSE estimate of the state given the
> observations (Thm. 2 in Kalman, 1960). The word *linear* was missing in the text, We will add this in the revision. Note that the error analysis also does not require independent noise.
>
> **Q:** *Also in the formulation of (5), what are random quantities and do we have any condition expectations or not?*
>
> **A:** The expectation in (5) is taken over all the random variables;
> consisting of the initial state $x_0$, dynamic noise $\xi(t)$, and
> measurement noise $\omega(t)$ for $t= 0, \cdots, T$. The conditional
> expectation is not necessary (the estimate is constrained to be
> measurable with respect to the history of observation).
>
> **Q:** It would be great if the results can be compared with more related works on learning the Kalman filter under different settings;
>
> **A:** Thanks for pointing us to the relevant references that we missed
> in our literature survey. We will include these references along with
> the summary of the following comparison in our revision.
>
> (Zheng, Y. et al. ) establishes an end-to-end sample
> complexity bound on learning a robust LQG controller--establishing a nice trade-off between optimality and robustness. As the system parameters are also unknown, this work only considers the *open-loop stable* systems. While our filtering design problem is based on the knowledge of system parameters, we do not require the open-loop stability
> assumption and its robustness analysis is part of our future work. Furthermore, the complexity
> bounds in (Zheng, Y. et al. ) depends on the length of trajectory and  scale as $O(1/\sqrt{N})$ in number of trajectories whereas ours does not depend on length and scales as $O(1/N)$.
>
> (Zhang, X. et al.) considers the problem of
> learning the steady-state Kalman gain but in a different setup: The
> model is assumed to be completely unknown. However, the algorithm
> requires access to a simulator that provides noisy measurement of the
> MSE  $\mathbb E[\|X(t)- \hat{X}(t)\|^2]$ which requires generation of
> ground-truth state trajectories $X(t)$ (see Assumption 3.2 in the
> reference). The proposed approaches are different: zeroth-order global
> optimization vs first-order stochastic gradient descent. Sample complexity result of our approach and their approach is similar. However, it is difficult to provide more detailed comparison as explicit dependence of the error terms on
> problem dimension is not provided.
>
> (Liu, W. et al) considers the problem of
> simultaneously learning the optimal Kalman filter and linear feedback
> control policy in LQG setup. Their approach involves solving SDP
> problems using input-state-output trajectories. Their result, for the
> the case when trajectories involve noise, relies on the assumption that
> the magnitude of the noise can be made arbitrary small. This is in
> contrast to our setup where we only assume a bound on the noise level
> and do not require access to state trajectories.

---

> > ### Comment · Reviewer_yyZM · 2023-08-17
> > **Thanks for the rebuttal!**
> >
> > Thanks for the effort in addressing my concerns. I have read the rebuttal. I am satisfied with the response and do not have further questions.

---

### Official Review · Reviewer_aTTt · 2023-07-11

**Soundness:** 3 good
**Presentation:** 3 good
**Contribution:** 2 fair
**Rating:** 4
**Confidence:** 3

**Summary:**

This paper focuses on learning the optimal filtering policy (Kalman gain) in a linear system with known system matrices and unknown covariance matrices of noise. The paper considers a proxy objective to avoid learning with hidden variables and characterizes a dual form of the objective, optimized subsequently using SGD. Convergence analysis and finite-time error bounds are provided.

**Strengths:**

originality: studies a traditional Kalman filtering problem with a different setting and objective. The duality theory and the proposed method of optimizing the dual objective seem original.

quality: provides detailed theoretical results and analysis.

clarity: clearly formulate the setting. The whole picture from the background of Kalman filtering, problem setting, duality theory, to SGD convegence analysis is well organized and easy to follow.

**Weaknesses:**

1. The paper lacks enough empirical experiments to support their idea. The only two figures appear in the supplementary materials, but the linear convergence outside a region near optimal, along with reasons why performances of the $M=20$ and $T=200$ case is different from other settings, is not presented clearly in this paper. It is suggested that:
 - the figures can be plot in the log scale to illustrate the linear convergence; also it is better to illustrate in the plot the scale of the "no-linear-rate" small region;
 - additional studies to demonstrate why medium hyperparameters $M=20$ and $T=200$, yield worst/best convergence rate among other hyperparameters (otherwise it is likely to think that the proposed SGD approach is not so robust to hyperparameters);
 - maybe comparison with some existing methods on real data is preferred.

2. The organization of this paper can be improved. For example,
 - numerical results can be put in main paper.
 - from my view, it is more natural to first characterize the bias in estimated gradient (Sec.4.2) and then provide convergence guarantees under such biased gradient (Sec.4.1). Otherwise, it may lead to confusion on the need of convergence analysis under biased estimates (e.g. the appeared $E$ in Lemma 5 & Proposition 2).

3. Some typos:
 - Eq.(3b) $P(T)$ should be $P(t).

**Questions:**

None

**Limitations:**

The main limitation of the proposed approach is that the biased gradient causes the convergence rate to be linear only outside a small region near the true optimal value. It may not seem so preferable if the region is large, thus not ensuring exact convergence to optimal value.

---

> ### Author Rebuttal · Authors · 2023-08-09
>
>
> The authors would like to express their gratitude to this reviewer for
> their insightful comments that helped us improve the
> presentation and empirical results of this manuscript.
>
> ***Q:** The paper lacks enough empirical experiments to support their
> idea. The only two figures appear in the supplementary materials, but
> the linear convergence outside a region near optimal, along with reasons
> why performances of the $M=20$ and $T=200$ case is different from other
> settings, is not presented clearly in this paper. It is suggested that:
> the figures can be plot in the log scale to illustrate the linear
> convergence; also it is better to illustrate in the plot the scale of
> the \"no-linear-rate\" small region; additional studies to demonstrate
> why medium hyperparameters and , yield worst/best convergence rate among
> other hyperparameters (otherwise it is likely to think that the proposed
> SGD approach is not so robust to hyperparameters); maybe comparison with
> some existing methods on real data is preferred.*
>
> **A:** Thanks for pointing out the strange nature of the empirical
> results which made us to look closely at the code and find a mistake in
> computing the exact steady-state Kalman gain (the multiplication by the
> $A$ matrix is not included in the code). The mistake produces an error
> of the order $A - I = O(\Delta t)$ which effected the relative order for
> different $M$. The corrected figures are produced in the accompanying
> pdf for the figures, which illustrates the expected order of the error
> curves and linear convergence regime (the errors are averaged over $50$
> simulations). Finally, please note that our contributions are mostly
> methodological and theoretical: the empirical results are provided for
> illustration purposes. Extensive numerical experiments for specific
> applications and comparison with related approaches is the subject of a
> separate work.
>
> ***Q:** The organization of this paper can be improved. For example,
> numerical results can be put in main paper. From my view, it is more
> natural to first characterize the bias in estimated gradient (Sec.4.2)
> and then provide convergence guarantees under such biased gradient
> (Sec.4.1). Otherwise, it may lead to confusion on the need of
> convergence analysis under biased estimates (e.g. the appeared in Lemma
> 5 & Proposition 2).*
>
> **A:** Thanks for the suggestion about the organization of the paper. We
> will move the numerical result to the main paper by moving supporting
> lemmas to the supplementary material. We will also interchange 4.1 and
> 4.2 as it fits better to the the flow of the arguments.
>
> ***Q:** The main limitation of the proposed approach is that the biased
> gradient causes the convergence rate to be linear only outside a small
> region near the true optimal value. It may not seem so preferable if the
> region is large, thus not ensuring exact convergence to optimal value.*
>
> **A:** We like to clarify that the convergence guarantee to a region
> around the optimal value is due to the finite-length of the data. The
> region can be made arbitrary small by choosing larger length and larger batch-size. In
> particular, to achieve $\varepsilon$ error, we only require the length
> $T\geq O(\ln(1/\varepsilon))$ and $M\geq O(1/\varepsilon)$. See the formal version of Theorem 1 in
> Response to All. Also, see the newly added Fig 1 (d) illustrating that
> optimality gap at the final iteration (i.e. the radius of the small
> neighborhood around optimality) is decaying linearly as a function of
> trajectory length $T \leq 50$---until the variance error dominates
> beyond $T=50$. It is clear that, increasing batch size $M$ will allow
> further decrease of this optimality gap beyond $T=50$.

---

### Official Review · Reviewer_LR36 · 2023-07-17

**Soundness:** 4 excellent
**Presentation:** 3 good
**Contribution:** 4 excellent
**Rating:** 7
**Confidence:** 4

**Summary:**

This paper studies the learning of optimal steady-state Kalman filter gain for a linear dynamical system with known system matrices but unknown process and measurement noise covariances. In particular, the learning process involves minimizing the prediction error of the observed output using a dataset of independently realized observation sequences. Leveraging the duality between LQR control and Kalman filtering, the authors reformulate this learning problem as synthesizing an optimal control policy for an adjoint system and propose a stochastic gradient descent (SGD) algorithm algorithm to solve it. The authors also provide sample complexity and non-asymptotic error guarantees by conducting a convergence analysis of SGD accounting for biased gradients and stability constraints. In particular, they show that the bias-variance error bounds scale logarithmically in system dimension and that the variance term doe snot change with the horizon.





**Strengths:**

I think this paper is well-written. The text is easy to read and and the  technical details are mostly neatly presented. The analysis also makes sense although I didn't verify in detail. The results are fairly general and significant as the problem of unknown noise covariances in Kalman filtering should be practically relevant as well.


**Weaknesses:**

I don't consider these as weaknesses necessarily but here are a few things I would recommend the authors to consider:
1. The results of remark 7 can be discussed earlier maybe within the informal theorem 1.
2. You might find it useful to adopt $(\kappa,\gamma)-$stability definition by Cohen et al. 2018 in your bounds involving $\sqrt{\rho(A_L)}$, especially in Lemma 6.
3.it wasn't clear to me in the first place why singularity of $H^TH$ requires a significantly different treatment than LQR case. maybe you can clarify this earlier in the text.
4. It would be helpful to discuss possible future directions.


**Questions:**

I have a few questions to the authors out of curiosity.  How essential is it to assume bounded noise for propositions 4 and 5? In figure 1a, could you explain why we see that M=10 case seems like the best performing , even better than M=30?



**Limitations:**

The authors already addressed some of the limitations such as the requirement for perfect knowledge of system matrices. I would also consider the bounded noise setting one of the limitations that might be improved in the future works.

---

> ### Author Rebuttal · Authors · 2023-08-09
>
> The authors would like express their gratitude to this reviewer for their insightful comments.
>
> **Q:** *The results of remark 7 can be discussed earlier maybe within the informal theorem 1.*
>
> **A:** We are going to replace Remark 7 with a formal version of Theorem 1 that includes the necessary assumptions and the complexity bounds (See Response to All). To avoid introducing symbols in the introduction, we keep the informal version of Theorem 1 the way it is and refer to its formal version for details to avoid complications.
>
> **Q:** *You might find it useful to adopt $(\kappa,\gamma)-$stability definition by Cohen et al. 2018 in your bounds involving $\sqrt{\rho(A_L)}$, especially in Lemma 6.*
>
> **A:** Thanks for pointing out the notation convention. Indeed, we can rewrite the bound in Lemma 6 in terms of $(\kappa,\gamma)$ as opposed to spectral radius. We will point this out in our revision. However, we decided to also keep the explicit  appearance of spectral radius in the bound to facilitate comparison with the related literature where spectral radius appears in the respective bounds (Fazel et. al. 2018, Tsiamis & Pappas, 2023).
>
> **Q:** *It wasn't clear to me in the first place why singularity of $H^\intercal H$ requires a significantly different treatment than LQR case. maybe you can clarify this earlier in the text*
>
> **A:**
> The existing convergence results for the LQR problem rely on the positive-definiteness of the covariance of the initial state. For instance, the constant $\mu = \lambda_{\min}(\Sigma)$ appears in all bounds in (Fazel et. al. 2018). We hint to this difference on line 51 of Introduction and provide more details in Remark 2 after we introduce the dual LQR problem in Proposition 1. We will modify Remark 2 to further clarify the difference with the existing analysis.
>
> **Q:** *It would be helpful to discuss possible future directions.*
>
> **A:** We will include the following directions for future research:
> - Single trajectory data: A direction for future research is to study how to adapt the proposed algorithm and error analysis for the setting when a single long trajectory is available as opposed to several independent trajectories of finite length.
> - In-accurate system matrices: An important research direction is to carry-out a robustness analysis, similar to its LQR dual counterpart, to study the effect of the error in system parameters on the policy learning accuracy.
> - Nonlinear setting:  The ultimate research goal is to use the recently introduced duality in nonlinear filtering (Kim. 2022) as a bridge to bring tools from RL to nonlinear filtering.
>
> **Q:** *How essential is it to assume bounded noise for propositions 4 and 5?*
>
> **A:** We are using standard Matrix Bernstein Inequality which requires almost sure norm-bounded realization of the random matrices. It is possible to improve this using more advanced results on concentration inequalities which is part of our future research direction.
>
> **Q:** *In figure 1a, could you explain why we see that $M=10$ case seems like the best performing , even better than $M=30$?*
>
> **A:** This was due to a mistake in our code. Please see the general response and the corrected figures in the accompanying pdf.

---

> > ### Comment · Reviewer_LR36 · 2023-08-21
> >
> > Dear authors, Thank you very much for the time and effort you spend to provide a clear and detailed response to my concerns and questions. I'm convinced by your answers and in favor of maintaining my score.

---

### Author Rebuttal · Authors · 2023-08-09

We appreciate reviewers' feedback and suggestions that have helped improve the paper. We provide a summary of the response to the main concerns here, followed by individual responses to the reviewers.

**Presentation of the technical results:** In order to improve the presentation of the technical result, we will make the following changes to the paper:

- Remark 7 will be replaced with the formal statement of Theorem 1. This will include the main result and the necessary assumptions:

**Theorem 1.** *Consider the linear system (1) under Assumptions 2 and 4. Suppose the SGD algorithm is implemented  with initial stabilizing gain $L_0$, the step-size $\bar\eta \coloneqq \frac{2}{9 \ell(J(L_0))}$, for $k$ iterations,  a batch-size of $M$, and data-length $T$. Then, $\forall \varepsilon>0$ and with probability larger than $1-\delta$, $J(L_k)  - J(L^\*)\leq \varepsilon$ if $$T \geq  O(\ln(\frac{1}{\varepsilon})),\quad  M \geq  O(\frac{1}{ \varepsilon}\ln(\frac{1}{\delta}) \ln(\ln(\frac{1}{\varepsilon}))), \quad \text{and}\quad k \geq O(\ln(\frac{1}{\varepsilon})).$$*

- In the beginning of Section 4, we will give a roadmap to navigate the technical results that concludes the proof of Theorem 1:
1. Section 4.1 is concerned with the convergence analysis of the SGD algorithm under the assumption that oracle gives a gradient estimate that satisfies $\|\nabla\widehat J(L) - \nabla J(L)\|_F \leq s \|\nabla J(L)\|_F + s_0$ whenever $\nabla J(L) \ge \frac{s_0}{\tau}$ for $s,\tau \in (0,1)$, and $s_0>0$. The main result of this section is Thm.2 which concludes the linear convergence of the iterates for sufficiently small $s_0$, i.e.
$J(L_k)-J(L^\*)\leq \varepsilon$, if $s_0\leq O(\sqrt{\varepsilon})$ and $k>O(\ln(\frac{1}{\varepsilon}))$.

2. Section 4.2 is concerned with the bias-variance error analysis of the gradient estimate, summarized as the main result in Thm. 3. It gives the sufficient values for the batch-size $M$ and trajectory length $T$ that provides the desired bound on the gradient estimate error required in Thm.2 with arbitrary small $s$ and $s_0$.

3. Combining the results from Thm. 2 and Thm. 3 concludes the main result Thm. 1. A proof sketch is provided at the end of this response.

**Explanation of the empirical result:** Thanks for pointing out the strange nature of the empirical results which made us reexamine the code and find a mistake in computing the exact steady-state Kalman gain (the multiplication by the $A$ matrix was not included in the code). The mistake produces an error of the order $A - I = O(\Delta t)$ which effected the relative order for different $M$. The corrected figures are produced in the accompanying pdf, which illustrates the expected order of the error curves and linear convergence regime. We also provide additional figures for the error at the final iteration, as a function of batch-size $M$ and time-horizon $T$. The numerical results serve to illustrate and validate the presented theoretical results. We leave extensive numerical experiments with real data on applications and empirical comparison with existing approaches for a separate work.

**limitations of the learning setup:** We agree that the perfect knowledge of the system matrices is a strong assumption. Our perspective is to separate the two problems of  identification of system matrices and the learning of the Kalman gain. This is aligned with certain practical considerations as we explain below. The system identification procedure  occurs through the application of physical principles and collection of data from experiments in a controlled environment (e.g., in a wind tunnel). However, identifying the noise covariance matrices strongly depends on the operating environment which might be significantly different than the experimental setup. Therefore, it is common in engineering applications to use the learned system matrices and tune the Kalman gain to improve the estimation error. Please see [Ref 1] for the application of this procedure for gust load alleviation on aircraft wings and [Ref 2] for estimation in chemical reactor models.  We also emphasize that the assumed learning setup has a rich history in adaptive filtering with numerous references with a  recent survey on this topic [Ref 3]. Our plan for future research is to carry-out a robustness analysis, similar to  its LQR dual counterpart in [Ref 4, Ref 5], to study the effect of the error in system parameters on the learning accuracy.

[Ref 1] Hinson KA, Morgansen KA, Livne E, ``Autocovariance least squares noise covariance estimation for a gust load alleviation test-bed,'' AIAA SCITECH 2022 Forum, 2021.

[Ref 2] Odelson BJ, Lutz A, Rawlings JB. ``The autocovariance least-squares method for estimating covariances: application to model-based control of chemical reactors,'' IEEE Trans. Control Syst. Technol., 2006

[Ref 3] L. Zhang, D. Sidoti, A. Bienkowski, K. R. Pattipati, Y. Bar-Shalom, and D. L. Kleinman,
391 ``On the identification of noise covariances and adaptive Kalman filtering: A new look at a 50
392 year-old problem,'' IEEE Access, vol. 8, pp. 59362–59388, 2020

[Ref 4] Safonov, M. G., and W. E. I. Z. H. E. N. G. Wang. ``Singular value properties of LQ regulators,'' IEEE transactions on automatic control 37.8 (1992): 1210-1211.

[Ref 5] Chen, Ci, and Anthony Holohan. ``Stability robustness of linear quadratic regulators.'' International Journal of Robust and Nonlinear Control 26.9 (2016): 1817-1824.

*Formal Proof of Theorem 1.* According to Thm.3, Assumption 3 holds at each iteration with probability at least $1-\delta$, if  $T \geq O(\ln(1/s_0))$ and $M \geq O(\ln(1/\delta)\big/ s_0^2)$. As a result, Thm.2 is valid, hence
$J(L_k)-J(L^\*)\leq \varepsilon$, if $s< \frac{1}{4}$, $s_0\leq O(\sqrt{\varepsilon})$ and $k>O(\ln(\frac{1}{\varepsilon}))$.
 Finally, the claim follows by combining the above bounds and using union bound for computing the failure probability of $k$ iterations.◻

---

### Decision · Program_Chairs · 2023-09-21

**Decision:**

Accept (poster)

**Comment:**

The authors consider the problem of learning the optimal (Kalman) filter for a linear system with unknown noise covariance matrices, giving a convergence analysis for SGD with bias-variance error bounds logarithmic in dimension. In discussion the authors justified the practicality of the problem setup. Reviewers found the contributions technically interesting and novel. The only reviewer giving a low rating has not acknowledged the author rebuttal, and is mainly concerned with experiments; the authors have adjusted their experiments in response, and besides, the contribution is mainly theoretical. I recommend acceptance.